# A comparison of DNA methylation detection between HiFi sequencing and whole genome bisulfite sequencing in monozygotic twins with Down syndrome

Kanyanee Promsawan[1], Chalurmpon Srichomthong[2,3], Monnat Pongpanich [4,5]*, Vorasuk Shotelersuk [2,3]

**1** Program in Bioinformatics and Computational Biology, Graduate School, Chulalongkorn University, Bangkok, Thailand, **2** Department of Pediatrics, Faculty of Medicine, Center of Excellence for Medical Genomics, Chulalongkorn University, Bangkok, Thailand, **3** Excellence Center for Genomics and Precision Medicine, King Chulalongkorn Memorial Hospital, the Thai Red Cross Society, Bangkok, Thailand, **4** Department of Mathematics and Computer Science, Faculty of Science, Chulalongkorn University, Bangkok, Thailand, **5** Center of Excellence for Cancer and Inflammation, Chulalongkorn University, Bangkok, Thailand

* Monnat.p@chula.ac.th

## Abstract

DNA methylation, a key epigenetic modification, regulates gene expression and diverse cellular functions. Bisulfite sequencing (BS) remains the gold standard for methylation detection, while PacBio HiFi sequencing enables direct detection without chemical conversion. Although both technologies are increasingly used, few studies have directly compared their concordance, particularly in clinically relevant settings such as Down syndrome (DS). We performed a comparative analysis of DNA methylation profiles using whole-genome bisulfite sequencing (WGBS) and PacBio high-fidelity (HiFi) whole-genome sequencing (WGS) in a pair of monozygotic twins with DS. WGBS data were processed with two pipelines, wg-blimp and Bismark, while HiFi WGS data were analyzed using pb-CpG-tools. Our analysis focused on four key aspects: CpG site detection, genomic distribution of methylated CpGs (mCs), average methylation levels, and inter-platform concordance. HiFi WGS detected a greater number of mCs—particularly in repetitive elements and regions with low WGBS coverage—while WGBS reported higher average methylation levels than HiFi WGS. Both platforms exhibited methylation patterns consistent with known biological principles, such as low methylation in CpG islands, and the relative methylation patterns across genomic features were largely concordant. Pearson correlation coefficients indicated strong agreement between platforms ($r \approx 0.8$), with higher concordance in GC-rich regions and at increased sequencing depths. Depth-matched comparisons and site-level down-sampling revealed that methylation concordance improves with increasing coverage, with stronger agreement observed beyond $20\times$. Our findings support the reliability of HiFi WGS for methylation detection and highlight

**Data availability statement:** Data cannot be shared publicly due to the privacy policy of the Center of Excellence for Medical Genomics, Faculty of Medicine, Chulalongkorn University. However, data are available from the Center of Excellence for Medical Genomics Data Access / Ethics Committee for researchers who meet the criteria for access to confidential data. Requests for data access may be directed to: Ms. Kanyanut Wongkanta Secretary, Center of Excellence for Medical Genomics Faculty of Medicine, Chulalongkorn University Email: kanyanut.w@redcross.or.th To ensure persistent and long-term availability, the data is securely stored on servers managed by the Center of Excellence for Medical Genomics, Faculty of Medicine, Chulalongkorn University, with regular backups and archival procedures in place.

**Funding:** VS received funding from the Health Systems Research Institute under grant number 67-095. The funders had no role in the study design, data collection and analysis, decision to publish, or preparation of the manuscript. For more information about the funder, visit https://www.hsri.or.th.

**Competing interests:** The authors have declared that no competing interests exist.

its advantages in regions that are challenging for bisulfite-based methods. This study demonstrates that HiFi WGS can serve as a robust alternative for genome-wide methylation profiling.

## Introduction

DNA methylation in the human genome is a pivotal epigenetic modification implicating the transfer of a methyl group (–CH3) to cytosine bases within CpG dinucleotides at position C5 to form 5-methylcytosine [1,2]. The mechanism plays a critical role in many cellular processes, e.g., transcription, chromosome stability, X chromosome inactivation, chromatin structure, genomic imprinting, and embryonic development though the regulation of gene expression [2,3]. Aberrant DNA methylation has been found to be related to many human diseases. Consequently, DNA methylation has become one of the most extensively studied areas in epigenetics and can serve as biomarkers for disease diagnosis and treatment.

For DNA methylation detection, high-throughput bisulfite genomic sequencing is regarded as a gold-standard technology and becoming an increasingly accessible technique [4,5]. It is a highly sensitive and effective method developed by Frommer and colleagues based on the conversion of genomic DNA by using sodium bisulfite. Cytosine residuals are converted to uracil while 5-methylcytosine residues are unaltered after the treatment of DNA with sodium bisulfite. This allows 5-methylcytosine to be discriminated from unmethylated cytosines [5]. Recently, there has been direct detection of DNA methylation without the need for bisulfite conversion and polymerase chain reaction (PCR) amplification by long-read sequencing technologies: Nanopore sequencing and PacBio HiFi Sequencing. Nanopore sequencing detects base modifications directly by measuring changes in electrical current. It predicts CpG methylation using hidden Markov models, neural networks, or statistical tests, depending on the selected workflow [6]. In PacBio HiFi sequencing, DNA methylation is measured based on the width and duration of the fluorescence pulses from the polymerase kinetic reaction [7]. This detection method uses a deep learning model which integrates sequencing kinetics and base context that provide high accuracy of methylation detection. It has also been shown that methylation detection with long-read sequencing is consistent with bisulfite sequencing [6,8,9]. However, previous studies comparing PacBio HiFi sequencing and bisulfite sequencing primarily focused on the average 5-methylcytosine (5-mC) rates across all samples. There are still limited studies on the concordance of 5-mC detection across various genomic regions, especially in complex genetic conditions like Down syndrome. While there is no biological rationale to expect that DS would affect the concordance between methylation platforms, the availability of short-read and long-read data provided an opportunity to extend the platform comparison to a disease setting, which has not been systematically explored.

Down syndrome (DS; also known as Trisomy 21) is recognized as the most typical chromosomal abnormality resulting from an extra copy of chromosome 21 [10]. DS

causes a complex variety of clinical symptoms, commonly characterized by intellectual disability and cognitive impairment [10]. Moreover, modifications including DNA methylation alterations may contribute to diverse disease phenotypes in DS. The literature has reported genome-wide epigenetic changes in DS that are associated with developmental impairments, including immune system dysfunction and brain development issues [11]. For DNA methylation detection in DS, extra genetic material can increase genomic complexity, which may lead to challenges in read mapping and methylation detection, particularly in repetitive regions. Furthermore, abnormal methylation patterns throughout the genome in DS, some of which may be low in frequency, could affect the overall accuracy of methylation state calls. Therefore, studying the concordance between different methylation technologies in this context is valuable.

Here, we aimed to compare DNA methylation (5-mC) analysis results between PacBio highly accurate long-read whole genome sequencing (HiFi WGS) and whole genome bisulfite sequencing (WGBS) data from a pair of Thai monozygotic twins with DS. Studying DNA methylation using monozygotic twins is uniquely advantageous, as they serve as well-matched controls for nearly all genetic variations and a wide range of environmental factors [12]. This design helps minimize cohort effects related to age, gender, genetic background, and early-life environmental exposures [13]. This study focused on assessing the concordance of CpG methylation predictions by stratifying the comparison across various genomic contexts, annotation categories, and sequencing depth levels. This comparative analysis was conducted to maximize insights gained from the existing data. Although the experiments were not initially designed for direct comparison, such as identical read depths, sample sizes or conditions, we leveraged the available data to evaluate cross-platform consistency in a disease context. To address differences in read depth between platforms, we performed down-sampling to match coverage at each CpG site. We investigated the distribution of methylated CpG (mC) sites across the genome to assess the consistency of both techniques in determining mC across various genomic regions. The regions were categorized into primary (sequence-based) and secondary (functional segment) levels. Primary levels include areas with varied CG densities, CpG-contexts (CpG islands, shores, and shelves), and repetitive regions (Tandem repeat, Long Interspersed Nuclear Elements (LINEs) and Short Interspersed Nuclear Elements (SINEs)). The secondary level encompasses gene-associated regions (promoters, exons, introns, untranslated regions (UTRs) and intergenic regions) and chromosomes. Additionally, we investigated the correlation between the methylation levels obtained from both methylation detection methods. This work contributes new insights into the consistency of HiFi WGS and WGBS in a genetically complex background and supports the applicability of long-read methylation detection in rare disease studies.

## Materials and methods

### Participants and sample collection

A pair of male monozygotic twins with trisomy 21 was recruited at 12 years and 9 months of age [14]. The weight and height of the twins at recruitment were 44 kg and 140.6 cm for Twin A, and 45 kg and 141.2 cm for Twin B, respectively. Genomic DNA was extracted from whole blood samples of both twins. The samples were collected with approval from the institutional review board of Faculty of Medicine of Chulalongkorn University (IRB number 264/62), with the overall recruitment period spanning from July 18, 2019, to July 17, 2025. While the present study focuses on a single pair of twins, the broader study continues to recruit additional participants. These twins were among the earliest participants enrolled in the study. Informed consent was obtained in writing from the parents of the twins, who were minor participants in this study.

### Methylation detection with WGBS and HiFi WGS

For WGBS, 10 µg of genomic DNA were sent to Macrogen, Inc. (Seoul, Korea) for library preparation with the Accel-NGS Methyl-Seq DNA Library Kit and sequencing with Illumina Platforms.

For HiFi WGS, 5 µg of genomic DNA were used for SMRTbell libraries using the SMRTbell Express Template Prep Kit 2.0 (P/N 100-938-900) (Pacific Biosciences, Menlo Park, CA). Incomplete or damaged SMRTbell molecules were

removed using SMRTbell Enzyme Clean-up Kit 2.0 (P/N 101-932-600) (Pacific Biosciences). Next, small DNA fragments (< 10 kilobases (kb)) were eliminated using BluePippin (Sage Science, Beverly, MA). Finally, the prepared SMRTbell libraries were sequenced on the Sequel II system that raw subreads were processed through the circular consensus sequencing (CCS) with kinetics workflow (PacBio SMRTLink version 10.0) to generate HiFi reads with a minimum estimated quality value (QV) of 20 (phred scaled, corresponding to an accuracy of 99 percent).

## Methylation detection analysis pipeline

WGBS data was analyzed to determine methylated CpGs (mCs) using the wg-blimp analysis pipeline v0.9.10 [15]. In brief, with the input of FASTQ files and the reference genome, sequence reads were aligned to the hg38 reference genome using Bwa-Meth [16] and deduplicated with picard [17]. After that, read quality was evaluated by FastQC [18] and Qualimap [19]. Then, methylation calling was performed by MethylDackel [20]. While wg-blimp was selected for its comprehensive and reliable WGBS workflow, Bismark v0.24.2 [21] was also used to validate the results and rule out tool-specific biases. In this approach, reads were aligned to a bisulfite-converted genome using Bismark with default settings, followed by deduplication and methylation extraction. Moreover, methylation levels in non-CpG contexts (CHG and CHH), as determined by Bismark [21] and MethylDackel [20], were analyzed to assess broader methylation patterns and bisulfite conversion efficiency. The estimated bisulfite conversion efficiency was calculated as: 100 – (% CHH methylation), with CHH methylation serving as the standard proxy for incomplete conversion [22,23].

HiFi WGS data was analyzed for mCs using the pb-CpG-tools v2.3.2 [24] in the following steps. First, HiFi reads with kinetics were generated from PacBio subreads BAM files by ccs PacBio [25], and sequence quality was examined using LongQC v 1.2.0 [26]. Next, CpG methylation annotation was performed by Jasmine v2.0.0 [27]. HiFi WGS reads with 5mc tags will be aligned to the hg38 reference genome using pbmm2 v1.9.0 [28]. Finally, mCs were investigated with the pb-CpG-tools v2.3.2 [24]. Mapping rates (Percent read mapped) for both analysis pipelines were calculated using the flagstat from Samtools v1.9 [29].

## Genomic context annotation

The location of each CpG site was annotated to multiple genomic contexts, enabling the examination of methylation patterns within different genomic features. Genome annotation files for different genomic contexts were downloaded from the UCSC Genome browser. For CpG context (CpG islands, shores, and shelves), the hg38 CpG island location file (https://hgdownload.soe.ucsc.edu/goldenPath/hg38/database/cpgIslandExt.txt) was downloaded to identify the position of CpG islands. CpG shore locations were constructed by adding 2 kb up and down from CpG islands with priority given to CpG islands in cases of overlap. CpG shelve regions were created from 2 to 4 kb from CpG island positions with priority given to both CpG islands and CpG shelves in instances of overlap.

For regions with different CG densities, the hg38 GC percent in 5-base windows (https://hgdownload.soe.ucsc.edu/goldenPath/hg38/bigZips/latest/hg38.gc5Base.wigVarStep.gz)was downloaded. When there is one C/G in a 5-base window, it results in 20 percent. If there are two C/G combinations in the 5-base window, it results in 40 percent, and so on. Therefore, GC percent tracks with 20, 40, 60, 80 and 100 percent were used in the study. These CG regions display the percentage of G (guanine) and C (cytosine) bases in 5 consecutive bases but are not constrained to "CG" dinucleotides.

Genomic elements comprising genes, transcripts, exons, coding sequence (CDSs) and UTRs were generated based on the positions from the GENCODE version 42 comprehensive gene annotation file. Promoters were constructed by expanding the region 2000 bp upstream from the transcription start sites (TSS). Exons of all transcripts were integrated, and introns were generated by subtracting exons from genes. Intergenic regions were constructed by deducting all other features (CDSs, promoters, genes) from the reference genome. The positions of regulatory elements, including open chromatin regions (DNase clusters) and enhancers (transcription factor binding clusters), were obtained from the UCSC Genome Browser annotation tracks.

DNA repetitive regions were circumscribed to simple repeat of tandem repeat expansion with repeating unit lengths from 1 to < 1000, Short Interspersed Nuclear Elements (SINEs) and Long Interspersed Nuclear Elements (LINEs). Tandem repeat positions were extracted from the simpleRepeats file (http://hgdownload.soe.ucsc.edu/goldenPath/hg38/database/simpleRepeat.txt.gz), an annotation generated by the Tandem Repeats Finder, while SINE and LINE regions were extracted from the RepeatMasker Track file (https://hgdownload.soe.ucsc.edu/goldenPath/hg38/database/rmsk.txt.gz), an annotation produced by the RepeatMasker program. Non-repetitive regions were generated by subtracting all repetitive regions identified in the RepeatMasker Track annotation and simple repeat annotation files from the genome.

### CpG distribution

The number and positions of methylated CpGs (mCs) in each twin, identified from WGBS and HiFi WGS methylation detection, were used to assess the distribution of mCs across the genome. CpGs were regarded as methylated if the methylation level was ≥ 50% and read coverage was ≥ 4 (default setting for read coverage). An alternative cutoff of ≥80% methylation was also considered to assess consistency across different threshold parameters. We divided mCs into 3 groups; 1) all mCs of each detection technique, 2) overlapping mCs between the two techniques, 3) mCs that were identified by only one technique. The distribution of mCs detected by both techniques was compared across various genomic contexts, i.e., primary (sequence) and secondary (functional segment) levels. Primary categories consist of sequencing regions with diverse CG densities, CpG-rich regions (including CpG islands, shores, and shelves), and repetitive regions (tandem repeats, SINEs and LINEs). The secondary levels include genetic regions (promoters, exons, introns, UTRs, and intergenic regions), regulatory elements (open chromatin and enhancer) and chromosomes. The number of mCs in different regions was normalized by dividing each count by the number of CpG sites in each region from the reference genome.

### Methylation level

Average methylation levels or methylation probabilities at overlapping CpG sites, predicted by MethylDackel [20] and Bismark [21] for WGBS data and pb-CpG-tools [24] for PacBio HiFi data, were compared across different genomic contexts. The correlation of methylation levels at these overlapping CpG sites between these two techniques were also determined by Pearson's correlation test.

### Gene-aligned methylation profiling

To profile methylation patterns across gene structure, we analyzed all protein-coding genes annotated in GENCODE v42, regardless of gene length. Because gene body lengths vary, we used a relative binning strategy in which each gene body was divided into 100 bins, with each bin representing 1% of that gene's total length. To capture surrounding regulatory regions, we additionally included fixed-length flanking regions: 2 kb upstream of the transcription start site (TSS) and 2 kb downstream of the transcription end site (TES), each divided into 20 equal-sized bins. This resulted in a total of 140 bins per gene: 20 upstream, 100 across the gene body, and 20 downstream. CpG methylation calls from PacBio HiFi and WGBS (processed via Bismark and wg-blimp with MethylDackel) were mapped to these bins using the GenomicRanges package in R [30], and average percent methylation per bin was visualized using ggplot2 [31].

### Depth-matched subsampling and concordance analysis

To control for coverage-related bias in methylation concordance analysis, we implemented a site-level depth-matched subsampling strategy. At each CpG site shared between platforms, the number of methylated and unmethylated reads was randomly subsampled from both HiFi WGS and WGBS data. The number of sampled reads was matched to the minimum read depth between the two platforms at that site, thereby ensuring a fair comparison while maintaining per-site

resolution and eliminating depth as a confounding factor. This procedure was repeated independently for 1,000 replicates. Pearson correlation coefficients (r) were calculated between HiFi WGS and WGBS methylation levels to evaluate concordance.

**Sequence entropy calculation**

To estimate sequence complexity in different genomic regions, we calculated DNA entropy using the Shannon entropy (H) formula [32]:

$$H = -\sum_{i=1}^{4} p_i log_2 p_i$$

where $p_i$ represents the frequency of each nucleotide (A, T, C, G) in the given region. For each category (tandem repeats, interspersed repeats, and non-repetitive regions), we computed the average entropy across all CpG-containing 100 bp windows overlapping those regions. These values provide a proxy for sequence complexity, with lower entropy indicating more repetitive sequences.

## Results

DNA methylation analysis was generated on 2 different sequencing data: WGBS and HiFi WGS data from a pair of monozygotic DS twins. For WGBS data, two analysis pipelines were used: wg-blimp and Bismark. The overall quality and characteristics of the sequencing data including average depth of coverage, number of reads, mean read length, GC content and percent read mapped were confirmed to be within the expected range before further analysis (S1 Table). Bisulfite conversion efficiency was consistently high (97.2–97.36%) across samples and pipelines, confirming effective conversion of unmethylated cytosines. Correspondingly, CpG methylation levels were high (83.7–85.4%), while non-CpG methylation remained low, with CHG and CHH methylation each accounting for only 2.4–2.8% (S2 Table). However, the sequencing depth between the two sequencing technologies may not have been optimally controlled due to the reliance on existing data.

For the comparative analysis, we examined four main aspects of methylation detection: first, the consistency in identifying number of CpG sites (methylated and unmethylated); second, the distribution of mCs; third, methylation probabilities or levels; and finally, the correlation between these methylation probabilities.

**The number of CpG sites**

Methylation detection analysis was determined using the wg-blimp [15] and Bismark [21] analysis pipeline for WGBS data, and pb-CpG-tools [24] for HiFi WGS data. HiFi WGS identified approximately 5.6 million more CpG sites (with total depth ≥ 4) than WGBS (using wg-blimp). CpG sites with methylation level at least 50 percent with a minimum read coverage of four were considered as mCs. HiFi WGS exhibits an increase of approximately 3.2 million mCs compared to WGBS. An overlap was observed between both methylation detection techniques, with almost 80 percent of the mCs identified in WGBS (using wg-blimp) also being present in methylation detection with HiFi WGS (Fig 1, S1 Table).

To confirm that our WGBS findings were not biased by the analysis pipeline, we reprocessed the WGBS data using Bismark. Although Bismark identified fewer total CpG sites and mCs with coverage ≥ 4 compared to wg-blimp, the overall methylation levels were consistent. Approximately 80% of mCs overlapped between the two pipelines, and Bismark also showed moderate overlap with mCs detected by HiFi WGS (S2 and S3 Tables).

To further assess CpG detection sensitivity, we analyzed CpG site counts across increasing minimum read coverage thresholds (4× to >60×) and plotted cumulative coverage distributions across three datasets: HiFi WGS, wg-blimp, and Bismark. The depth distribution for PacBio HiFi (S1A Fig) shows a unimodal and symmetric pattern peaking at 28–30×,

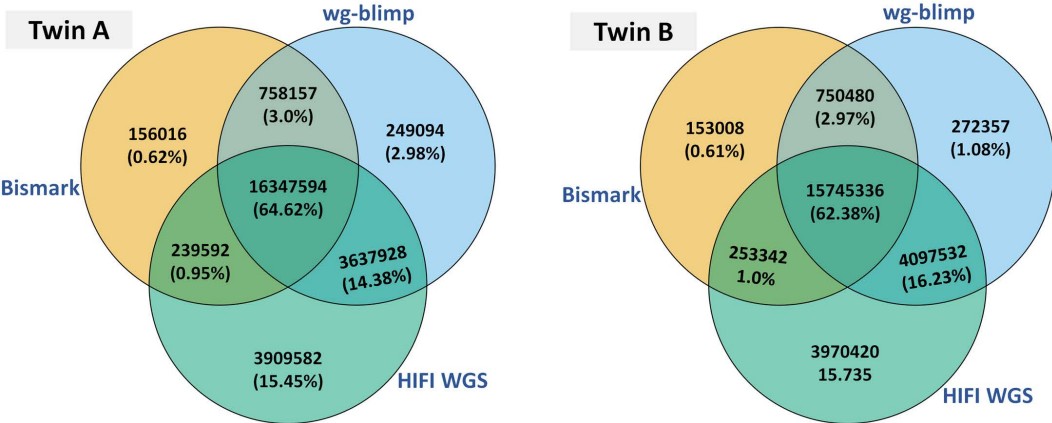

**Fig 1. Percentage of overlapping mCs.** Venn diagrams display the mCs that overlap between WGBS (wg-blimp), WGBS (Bismark) and HiFi WGS sequencing data of the twins.

indicating relatively uniform coverage. In contrast, both WGBS datasets (S1B,C Fig) display right-skewed distributions, with the majority of CpGs covered at low depth (4–10×) and relatively few achieving higher coverage. Over 90% of CpGs in the PacBio HiFi dataset have ≥10×coverage, compared to approximately 65% in the wg-blimp WGBS dataset and under 50% in the Bismark WGBS dataset, as estimated from the cumulative plots (S1 Fig). Notably, while most CpGs in the WGBS datasets are concentrated at lower depths, the final bin (>60×) includes a small number of CpG sites with extremely high coverage (some exceeding 4000×), which inflate the average coverage values reported in the summary tables and explain the discrepancy between the mean and the more typical coverage levels observed.

## The distribution of methylated CpG dinucleotides (mCs)

The distribution of mCs across the genome in different genomic contexts was investigated to assess the consistency of CpG predicted positions between the two sequencing technologies. The genomic contexts were categorized into two levels: primary and secondary levels. We examined mCs using three different procedures: first, by considering all mCs identified by each detection technique; second, by focusing on CpG sites that overlapped between the two techniques; and third, by isolating CpG sites that were unique to only one technique. Overall, HiFi WGS detected a greater number of mCs while maintaining broadly consistent methylation patterns with WGBS results generated by both the wg-blimp and Bismark pipelines in both twins (Figs 2–3, S2 and S3 Figs). While the overall trends were preserved across methods—for example, increasing methylation levels from CpG islands to shores and shelves—there were noticeable differences in specific genomic contexts. In particular, methylation profiles between HiFi WGS and Bismark showed greater divergence compared to those between HiFi WGS and wg-blimp. For example, in repetitive elements, HiFi WGS showed higher methylation in SINEs than in LINEs, whereas Bismark displayed the opposite trend (Figs 2 and 3, S2 and S3 Figs). Between the two WGBS analysis pipelines, wg-blimp consistently reported a higher number of mCs than Bismark, yet both showed largely concordant patterns across genomic features (S4 and S5 Figs). To further assess the robustness of our methylation threshold, we applied a more stringent cutoff of 80%. The results remained consistent, with over 80% of mCs overlapping those identified using the original 50% threshold (S4 Table). Overall methylation patterns remained similar across comparisons—HiFi WGS vs. wg-blimp, HiFi WGS vs. Bismark, and wg-blimp vs. Bismark— with the proportion of mCs uniformly decreasing by approximately 10–15% across different genomic regions for all three methods (S6–S11 Figs). The comparison between the 50% and 80% thresholds also showed consistent trends across HiFi WGS, wg-blimp, and Bismark in most genomic contexts. An exception was observed in CG density categories, where the HiFi WGS profile

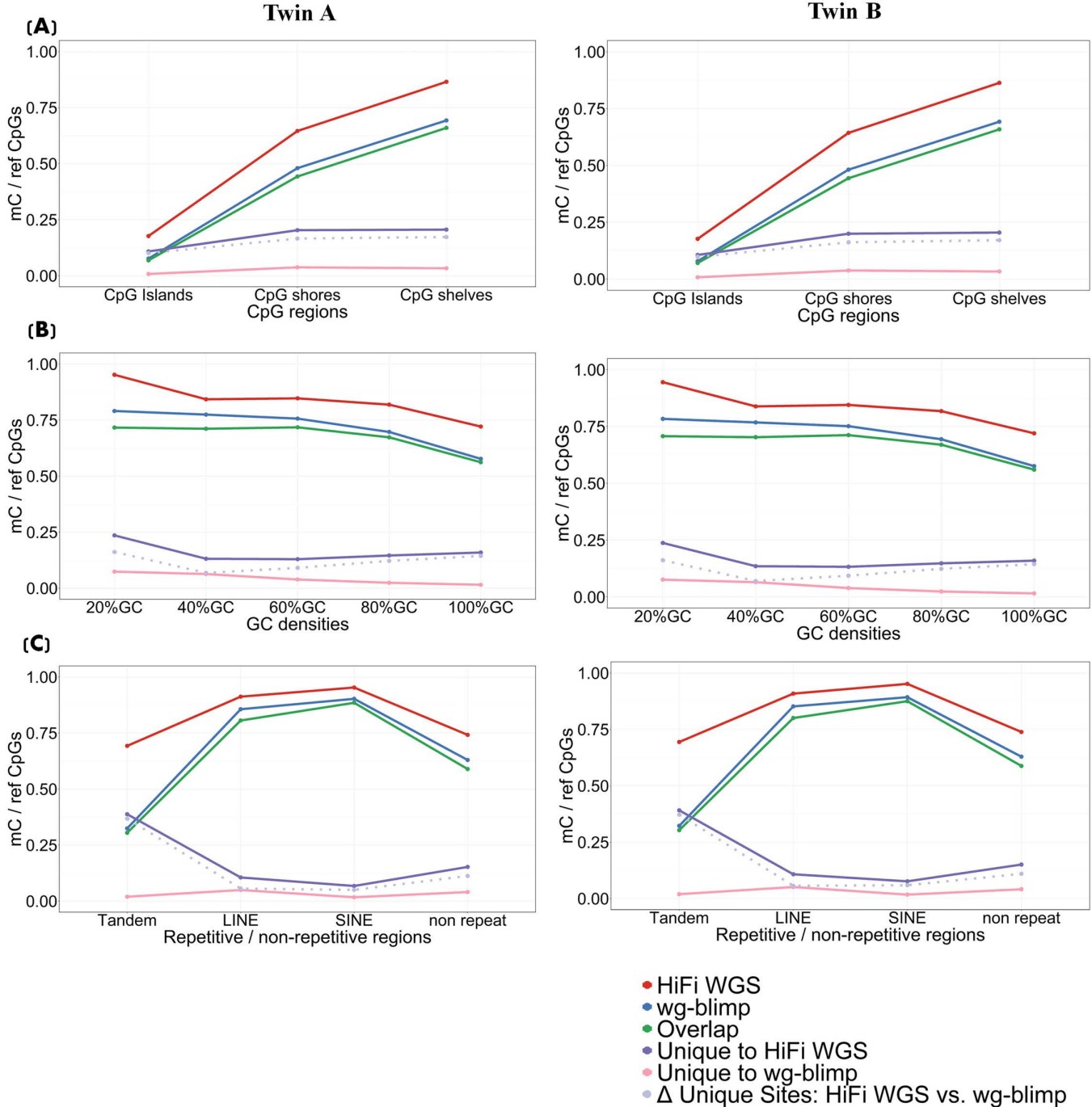

**Fig 2. Distribution of mCs across primary (sequence-level) genomic contexts in HiFi WGS and WGBS (wg-blimp).** Proportions of mCs (defined as ≥50% methylation with ≥4 × read coverage) are shown across sequence-based features: (A) CpG regions (islands, shores, and shelves), (B) CG density categories, and (C) repetitive elements. Data are shown for WGBS, HiFi WGS, overlapping mCs (Overlap), uniquely identified mCs in WGBS (Unique to WGBS), uniquely identified in HiFi WGS (Unique to HiFi WGS), and the difference between the unique sets (Δ unique sites: HiFi WGS vs. WGBS).

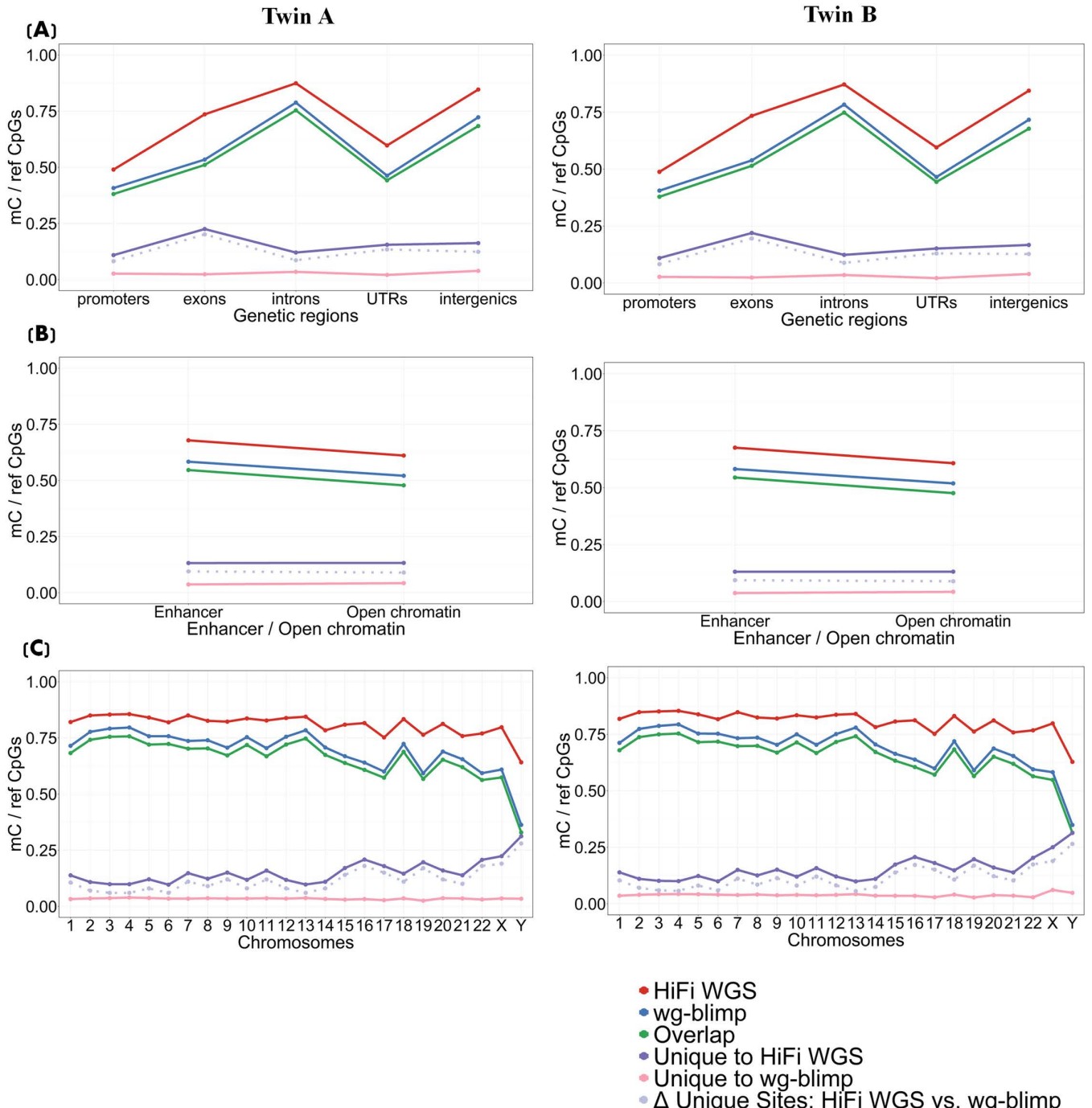

**Fig 3. Distribution of mCs across secondary (functional level) genomic contexts in HiFi WGS and WGBS (wg-blimp).** Methylated CpG proportions (≥50% methylation with ≥4 × read coverage) by: (A) gene-associated regions, (B) regulatory elements (open chromatin and enhancers), and (C) chromosomes. Data are presented for WGBS, HiFi WGS, Overlap, Unique to WGBS, Unique to HiFi WGS, and Δ unique sites.

shifted from a clear decreasing trend under the 50% threshold to a more variable, non-monotonic pattern under the 80% threshold (Figs 2 and 3, S2–S11 Figs).

When considering mCs in primary levels (GC-dense regions, regions with different GC densities, and repetitive regions), HiFi WGS identified a greater count of mCs across all regions. The distribution pattern mostly remained consistent between PacBio and WGBS (via both wg-blimp and Bismark) data in both twins and compatible with conventional biological principles (Fig 2 for HiFi WGS vs. wg-blimp; S2 Fig for HiFi WGS vs. Bismark). Among the GC-dense regions, CpG islands, which are GC-rich areas with a GC content higher than 50 percent (average of about 60 percent), demonstrated limited mCs, whereas CpG shelves show the highest proportion of mCs (Fig 2A; S2A Fig). Additionally, the mCs were tallied in regions with different GC densities, aiming to assess whether GC density impacts the effectiveness of 5mC predictions obtained through WGBS and HiFi WGS. In this study, GC density with 20, 40, 60, 80 and 100 percent were identified. GC density is derived from the percentage of G and C bases present in 5-base windows, and high GC content is usually linked to regions abundant in genes. The outcomes from both methylation detection methods appear to be in concordance for both thresholds (50% and 80%). The regions with 100 percent GC content had the lowest mCs (Fig 2B; S2B, S4B, S6B, S8B, and S10B Figs). Among the repetitive regions, Interspersed repeats (LINEs and SINEs) showed a higher proportion of mCs compared to non-repetitive regions, while tandem repeats exhibited the lowest levels of mCs. Moreover, there were more mCs specific to only HiFi WGS analysis, but not detected from WGBS in tandem repeats (Fig 2C; S2C, S4C, S6C, S8C, and S10C Figs). Notably, methylation patterns from the primary levels were consistently observed across both WGBS pipelines (S4 Fig).

The distribution patterns in secondary regions were largely consistent between the two techniques for both thresholds (50% and 80%) across various genetic regions including regulatory regions (enhancer and open chromatin) but exhibited variation across certain chromosomes (Fig 3; S3, S5, S7, S9, S11 Figs). In genetic regions, the lowest proportion of mCs were remarkably indicated in promoters. HiFi WGS remarkably identified more mCs in exons than WGBS (Fig 3A; S3A, S5A, S7A, S9A, and S11A Figs). For regulatory regions, both technologies showed a consistent pattern in which enhancer regions had a slightly higher proportion of mCs compared to open chromatin regions (Fig 3B; S3B, S5B, S7B, S9B, and S11B Figs). Across all comparisons, the chromosomal distribution of mCs exhibited consistent patterns across detection methods and thresholds, with the exception of chromosomes 15–16, where HiFi WGS showed a slightly elevated proportion of mCs, diverging from the downward trend observed in both Bismark and wg-blimp (Fig 3C; S3C, S5C, S7C, S9C, and S11C Figs). HiFi WGS reported the highest mC proportions across all chromosomes, followed by wg-blimp and then Bismark, with all methods showing a pronounced reduction on chromosome Y. Wg-blimp and Bismark produced highly comparable mC levels and profiles across chromosomes. Notably, under the more stringent 80% threshold, the proportion of mCs declined uniformly across chromosomes for each method; however, the relative differences between platforms and the chromosome-specific patterns remained consistent with those observed at the 50% threshold. These findings demonstrate that despite varying detection thresholds and analytical pipelines, the chromosome-level methylation landscape is largely reproducible and biologically meaningful (Fig 3C; S3C, S5C, S7C, S9C, and S11C Figs).

While HiFi WGS detected a larger number of mCs across all regions, a notably high percentage of uniquely identified mCs by HiFi WGS was observed in specific regions in both primary (e.g., tandem repeats) and secondary (e.g., exons, chromosome Y) genomic contexts. After inspecting the mC positions that uniquely reported by HiFi WGS, we observed that in these positions there wasa depth of coverage less than 4 in WGBS (wg-blimp) data for approximately 80 percent, and nearly 7 percent had methylation levels below 50, not meeting the criteria for mCs (S12 Fig; column 1 and 2). Regions with larger differences in mC levels between the two methods, such as tandem repeats and chromosome Y, showed a higher prevalence of CpGs with a sequencing depth below 4 in the WGBS data (S13 Fig). Likewise, WGBS identified mCs that were not present in the results from HiFi WGS, although in a smaller number. We also examined the equivalent positions in the HiFi WGS data; approximately 97 percent showed methylation levels below 50, and the

remaining (3 percent) had a coverage depth below 4 or alternative variants (S12 Fig; column 3 and 4). Consequently, these positions were excluded from the list of mCs.

## Methylation level across the genome and different genomic contexts

The average methylation levels at overlapping CpG sites from two sequencing technologiesof the twins were compared to gauge their concordance. The average methylation levels analyzed across different genomic regions at both primary and secondary levels using WGBS and HiFi WGS exhibited a largely consistent trend and concordance in both twins, with WGBS showing a slightly higher level of methylation (Fig 4 and 5 for HiFi WGS vs. wg-blimp; S14, S15 Figs for HiFi WGS vs. Bismark). The average methylation probability for all positions was approximately 83 percent for WGBS and 80 percent for HiFi WGS. CpG Islands showed the least amount of methylation compared to other CpG contexts, which are CpG shores and CpG shelves (Fig 4A; S14A and S16A Figs; leftmost and middle columns). When examining methylation levels across GC density bins, HiFi WGS and WGBS (wg-blimp and Bismark) exhibited slightly distinct patterns. HiFi WGS showed a gradual increase in methylation level from 20% to 80% GC, with a small drop at the 100% GC bin. In contrast, wg-blimp and Bismark demonstrated a relatively flat profile, with only minimal increases across GC bins and a comparable dip at 100% GC. These results highlight a moderate divergence in GC-dependent methylation quantification between the two technologies (Fig 4B; S14B and S16B Figs; leftmost and middle column). Furthermore, repetitive regions (Tandem

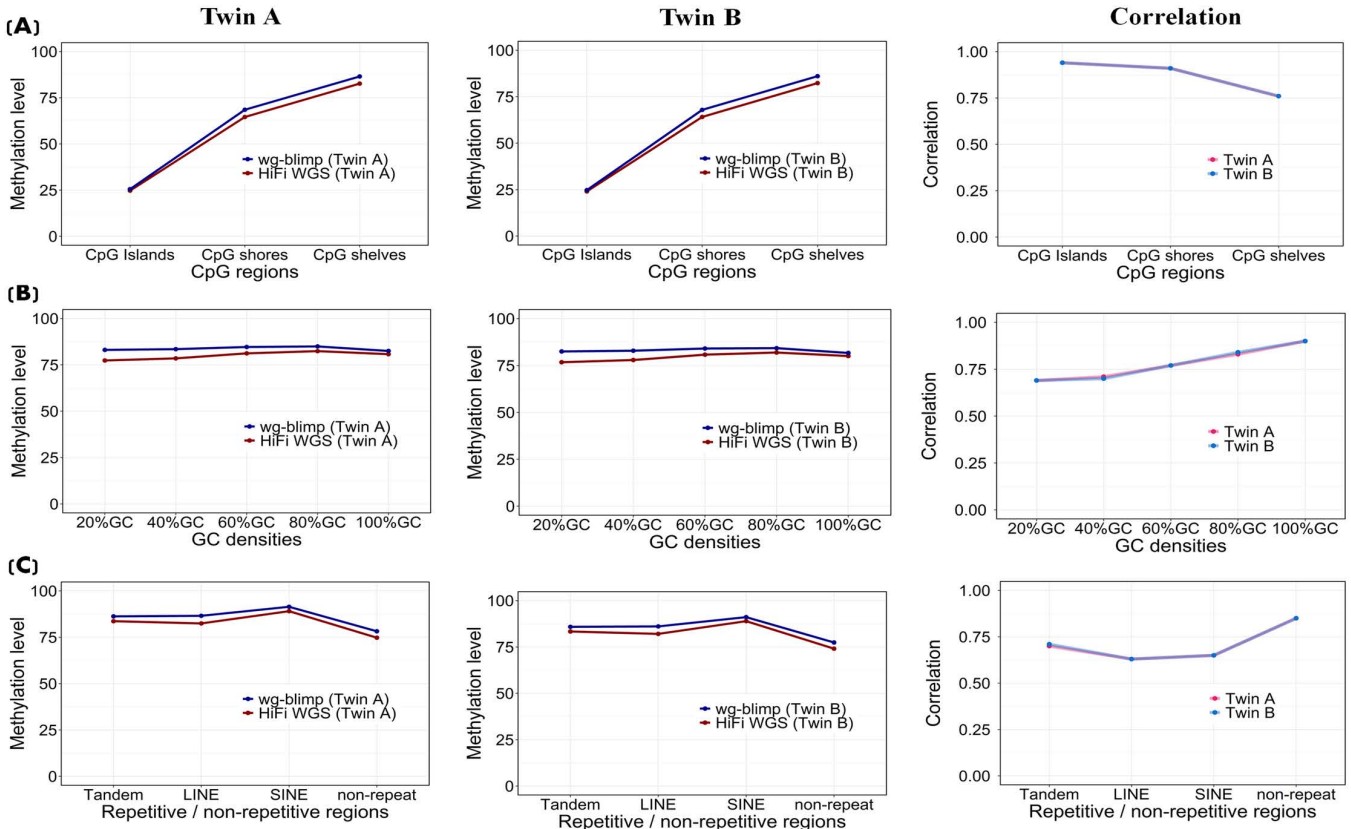

**Fig 4. Methylation levels and correlation across primary (sequence-level) genomic contexts in HiFi WGS and WGBS (wg-blimp).** Methylation levels (methylation probabilities) and Pearson correlation between WGBS and HiFi WGS across: (A) CpG regions (CpG islands, shores, and shelves), (B) CG density categories, and (C) repetitive elements.

repeats, LINEs and SINEs) displayed a slightly higher level of methylation compared to non-repetitive regions (Fig 4C; S14C and S16C Figs; leftmost and middle column). Among different genomic features, promoters, which are often located within or near CpG islands, illustrated the lowest level of methylation probability (Fig 5A; S15A and S17A Figs leftmost and middle column). Both enhancer and open chromatin regions exhibited moderately high methylation levels, with a slight reduction observed in open chromatin (Fig 5B; S15B and S17B Figs; leftmost and middle column). The average methylation pattern from 2 technologies were consistent across different chromosomes, with the lowest probability observed on chromosome Y. Wg-blimp and Bismark consistently reported slightly higher methylation levels than HiFi WGS by a small margin (~1–2%), but the relative pattern across chromosomes was preserved. This suggests that the overall methylation landscape is reproducible between methods at the chromosome scale (Fig 5C; S15C and S17C Figs; leftmost and middle column). Moreover, the methylation levels obtained from wg-blimp and Bismark were nearly identical across genomic regions, indicating strong consistency between pipelines (S16 and S17 Figs; leftmost and middle column). Overall, methylation levels followed expected biological trends.

Interestingly, when comparing methylation levels to the proportion of mCs shown in Figs 2 and 3, the two measurements generally followed similar trends—regions with a higher proportion of mCs also tended to exhibit higher methylation levels. However, there were notable exceptions. For instance, in GC density categories (Fig 2), regions with lower GC content showed a higher proportion of methylated sites, whereas methylation levels remained relatively stable or increased slightly with higher GC content. Tandem repeats represented another exception: despite exhibiting high

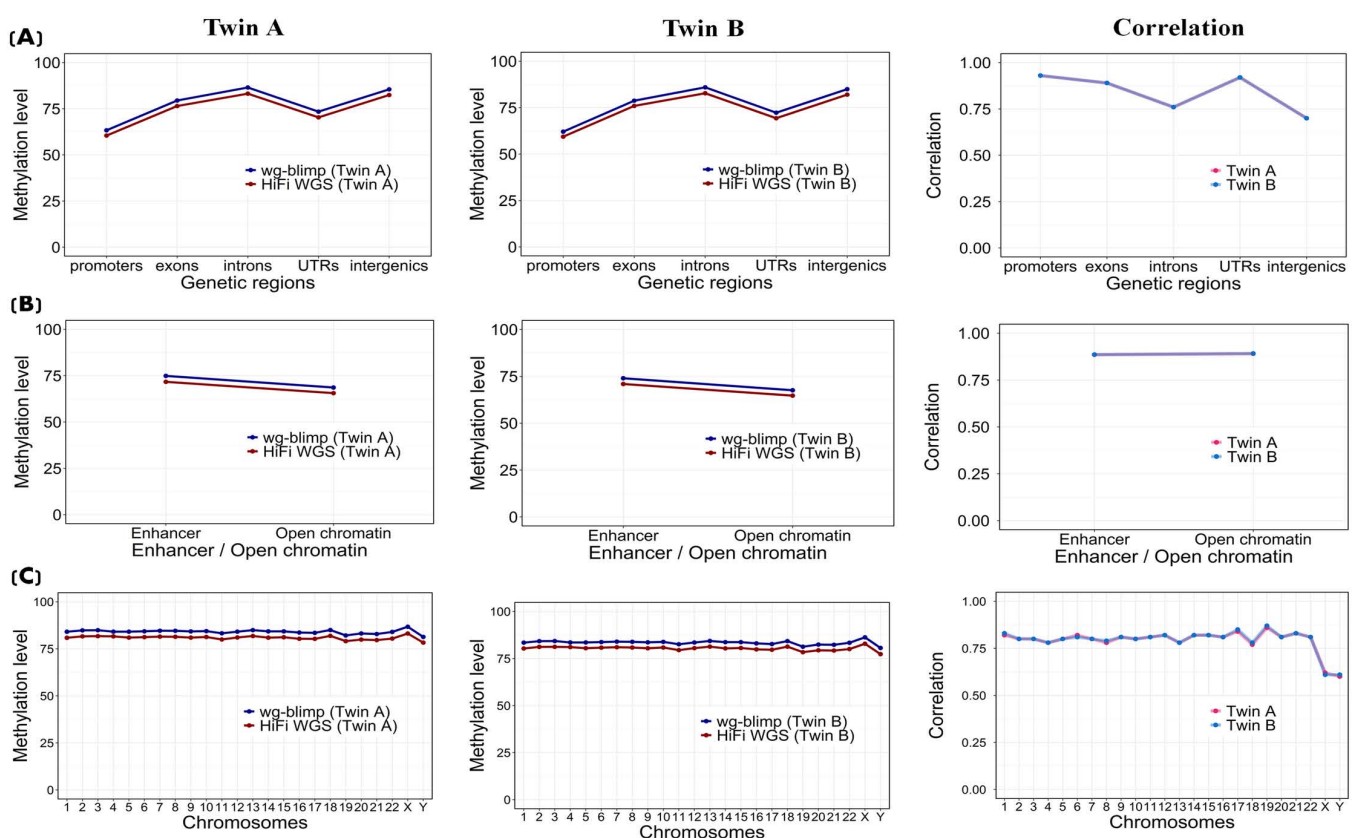

**Fig 5. Methylation levels and correlation across secondary (functional-level) genomic contexts in HiFi WGS and WGBS (wg-blimp).** Methylation levels and Pearson correlation between WGBS and HiFi WGS across: (A) gene-associated regions, (B) regulatory regions (open chromatin and enhancers), and (C) chromosomes.

methylation levels, they had a relatively low proportion of mCs. Additionally, while the number of mCs varied substantially across chromosomes, the average methylation level per chromosome remained relatively stable, further highlighting the distinction between methylation density and methylation level.

In addition, to examine methylation in relation to gene structure, average signal plots were generated spanning 2 kb upstream of the transcription start site (TSS), the scaled gene body, and 2 kb downstream of the transcription end site (TES). All platforms showed a characteristic dip in methylation near the TSS, elevated levels across the gene body, and a slight decline near the TES. The consistency across platforms (HiFi WGS, WGBS-Bismark, and WGBS-MethylDackel) supports the robustness of methylation measurements in genic and promoter regions (S18 Fig). We also investigated non-CpG methylation (CHG and CHH contexts), which showed consistently low levels across all genomic contexts (S19 Fig).

## Methylation probability correlation

Correlation of methylation levels at matched CpG sites between detection methods—HiFi WGS vs. WGBS (wg-blimp), HiFi WGS vs. WGBS (Bismark), and WGBS (wg-blimp) vs. WGBS (Bismark)—was assessed. Overall, the methods showed strong agreement, with Pearson correlation coefficients of 0.80 for HiFi WGS vs. wg-blimp, 0.73 for HiFi WGS vs. Bismark (lowest), and 0.93 for wg-blimp vs. Bismark (highest). Consistent patterns were observed between twins. In all comparisons, strong correlations were observed for CpG sites within CpG islands ($r > 0.92$), with gradually lower correlations in CpG shores and CpG shelves (Fig 4A; S14A and S16A Figs; rightmost column). CpG sites with higher densities of 80 and 100 percent exhibited stronger correlation compared to CpG sites with GC densities of 20, 40, and 60 percent (Fig 4B; S14B and S16B Figs; rightmost column). The correlation trends in CpG regions (Fig 4A; S14A and S16A Figs; rightmost column) and GC densities (Fig 4B; S14B and S16B Figs; rightmost column) align, with both showing highest concordance in CpG islands and high-GC regions, which often overlap in the genome. Moreover, the correlation of CpG sites in non-repetitive regions was higher than that in repetitive regions (Fig 4C; S14C and S16C Figs; rightmost column). Higher degree of correlation also identified in promoters, exons, and UTRs when compared to intron and intergenic regions (Fig 5A; S15A and S17A Figs; rightmost column). Correlation of methylation levels between detection methods was consistently high across both enhancer and open chromatin (Fig 5B; S15B and S17B Figs; rightmost column). Scatter plots (2D binned heatmaps) with linear regression lines showing concordance between WGBS (wg-blimp) and HiFi across different genetic contexts are shown in Fig 6 (Twin A) and S20 Fig (Twin B). Interestingly, it is worth highlighting that the correlation of methylation level in different contexts between methods tend to be higher in GC-rich regions (e.g., CpG Islands, promoter, region with high GC densities) when compared to GC- poor regions (e.g., intronic, and intergenic). Furthermore, chromosome-level correlation of methylation level was generally high across all autosomes, but noticeably lower on chromosomes X and Y (Fig 5C; S15C and S17C Figs; rightmost column).

We then explored whether the decreased correlation observed in specific regions at the primary level influences the correlation at the secondary level. The correlations tended to be higher in regions with high GC content. As a result, lower correlations were observed in secondary regions that predominantly consist of low GC density areas. For example, the proportion of introns and intergenic regions were lower in regions with higher GC densities (S21 Fig). The majority of CpG sites on chromosomes X and Y were from regions with lower GC densities and intergenic regions (S22 Fig).

To address potential coverage-related bias in methylation concordance, we compared HiFi WGS and WGBS using coverage-matched CpG sites. At lower coverage bins (5–20×), correlations were moderate ($r = 0.53$–$0.65$), whereas at higher depths (≥31×), concordance improved substantially ($r ≥ 0.72$). A decline in correlation was observed in the highest-depth bins (Fig 7 (Twin A) and S23 Fig (Twin B)). Although the scatterplots may appear noisy at first glance, this visual impression is primarily driven by a small number of discordant points, as indicated by dense clusters of blue-colored (low-frequency) dots. In contrast, the majority of data points are concentrated along the upper-right diagonal, highlighted by the yellow gradient, indicating strong agreement in methylation levels among coverage-matched CpGs. These results underscore the importance of sufficient sequencing depth for reliable methylation quantification. To further account for

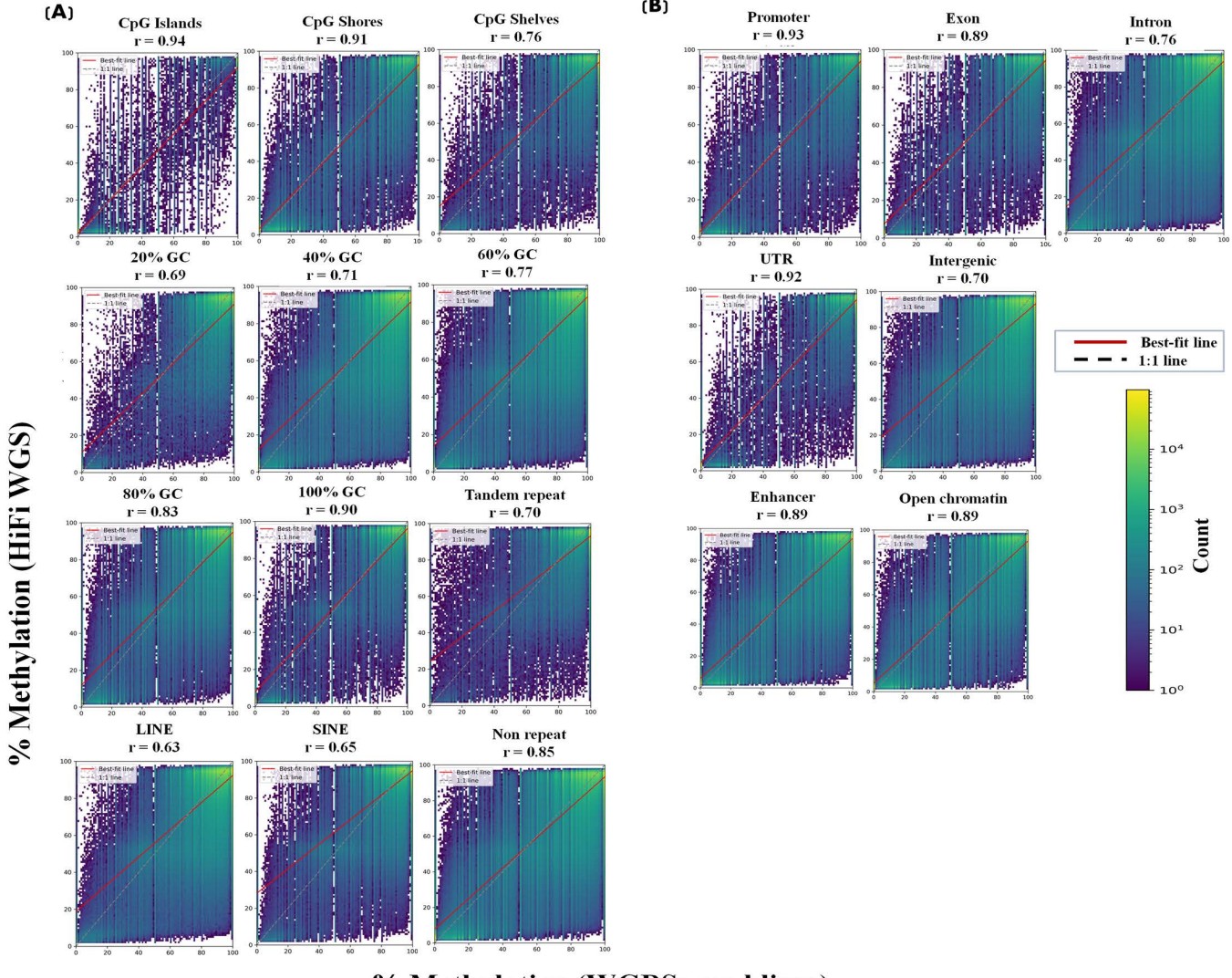

**Fig 6. Two-dimensional heatmaps with best-fit lines showing methylation concordance between WGBS (wg-blimp) and HiFi WGS across genomic contexts (Twin A).** Scatter plots (2D binned heatmaps) with linear regression lines illustrate methylation level concordance between WGBS and HiFi for CpG sites across: (A) CpG regions (islands, shores, and shelves), CG density categories, repetitive elements, (B) gene-associated regions, and regulatory regions (open chromatin and enhancers).

coverage-related bias, we implemented a site-level, depth-matched down-sampling strategy for the HiFi WGS and WGBS datasets. At each CpG site, methylation values from the higher-depth platform were randomly down-sampled to match the read depth of the lower-depth platform. This approach enabled the inclusion of a much larger number of CpG sites compared to equal-depth filtering (Fig 7; S23 Fig). To reduce stochastic variation from random sampling, we performed 1,000 rounds of down-sampling per CpG site and used the average methylation level for downstream analysis. Pearson correlation coefficients were then calculated between the down-sampled HiFi WGS and WGBS methylation levels across various genomic features. The results demonstrated good concordance, with correlation patterns (Fig 8) that closely resembled those observed in the original (non-downsampled) data (Figs 4, 5). To further examine the influence of

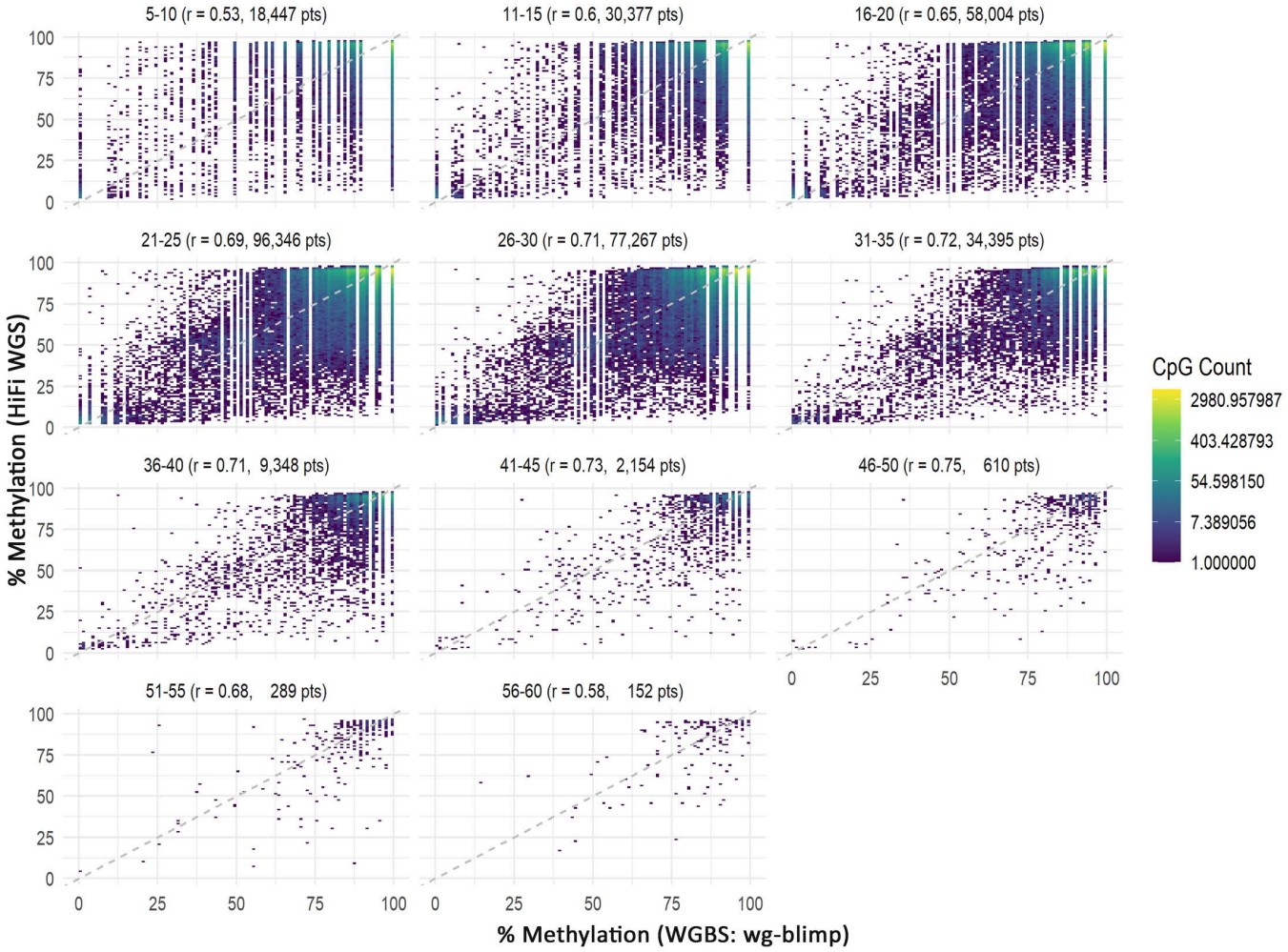

**Fig 7. Two-dimensional binned heatmaps of coverage-dependent methylation concordance between WGBS (wg-blimp) and HiFi WGS.**
Heatmaps show the correlation of methylation levels from WGBS (x-axis) and HiFi WGS (y-axis) across CpG sites, stratified by WGBS depth bins (e.g., 5–10×, 11–15×). Only CpG sites with matched coverage in both platforms are included. Color intensity reflects the density of CpG sites within each methylation bin.

sequencing depth on methylation concordance, we visualized the distribution of Pearson correlation coefficients across 1,000 down-sampling replicates at each PacBio depth level (S24 Fig). This analysis provides a complementary view to Fig 8 and reveals a non-linear relationship between depth and concordance. Specifically, correlation values initially decline as depth increases from 4× to approximately 20×, then gradually rise and reach a peak at 46×. Beyond this point, correlation values decline again.

## Discussion

In this study, we performed a comparison of CpG methylation analysis across genome to examine the agreement in number of CpG sites, mCs distribution, methylation probability (methylation level) and methylation probability correlation between the two sequencing technologies (HiFi WGS and WGBS) using the data from monozygotic twins with down syndrome. For WGBS, Bismark was used alongside wg-blimp, and bisulfite conversion efficiency was consistently high,

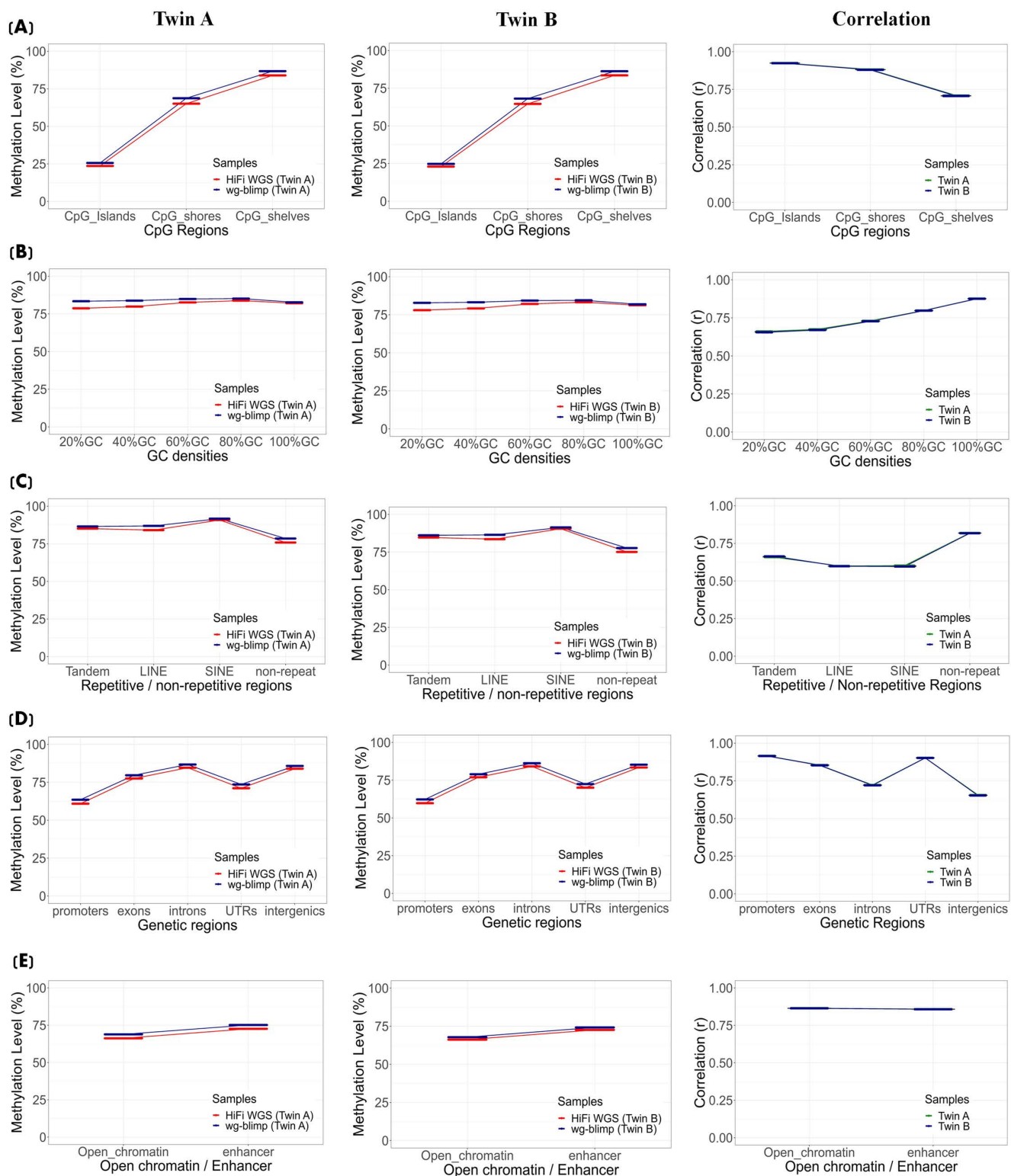

**Fig 8. Methylation concordance between HiFi WGS and WGBS across genomic contexts after depth-matched downsampling.** Methylated and unmethylated counts were randomly downsampled at each CpG site to the lower read depth between platforms to ensure matched coverage for fair comparison. Pearson correlation coefficients (r) were calculated across 1,000 iterations. Boxplots represent the distribution of r-values across various genomic contexts.

supporting the reliability of the methylation calls. From the result, consistent patterns were observed in the distribution of mCs and methylation probabilities across primary and secondary regions in both methods. There was an agreement in methylation detection patterns with an overall correlation coefficient for methylation levels of approximately 0.8. Even so, HiFi WGS detects a larger number of mCs, while WGBS shows higher overall methylation levels across the genome. Moreover, wg-blimp and Bismark from WGBS data show concordance in methylation pattern with very high correlation (r > 0.9) across genome.

The distribution of mCs showed consistent patterns between the two approaches in both primary and secondary regions. Therefore, there were no differences in the primary regions that influenced methylation patterns at the secondary level. However, a larger number of mCs were identified by HiFi WGS in all regions. Most of the missing mCs in the WGBS data were due to low coverage depth (< 4). The sequencing depth may not have been optimally controlled since we utilized existing data, leading to lower depth (< 4) in some CpG sites in WGBS compared to HiFi WGS. Moreover, Bismark tends to report lower CpG depths than wg-blimp, which may be due to differences in alignment stringency, read processing, and CpG counting strategies.

Moreover, there are differences in the methylation detection algorithms used between HiFi WGS and WGBS. The default settings of pb-CpG-tools for HiFi WGS enable de novo DNA methylation analysis, reporting CG sites beyond reference sequences. This disparity can account for the differing numbers of CpGs detected by the two technologies; hence, higher number of mCs in HiFi WGS. Moreover, the previous study suggested that variations in the DNA sequence near or within CpG sites can introduce mapping biases in WGBS, affecting the accuracy of methylation measurements [9].

Even though in a smaller number, WGBS identified mCs that were not present in the results from PacBio HiFi. Most of the absent positions (97%) exhibited methylation levels below 50. This indicates that two methods show discrepancies in predicting mCs at certain positions, albeit with a very small frequency (~1 percent).

Average methylation levels in all regions are higher in WGBS. Apart from the different algorithms used for DNA methylation detection between the two technologies, it can be inferred that the bisulfite conversion process may cause DNA degradation, potentially resulting in the loss of reads with unmethylated cytosines [33]. The incomplete BS conversion is widely recognized as critical since it can lead to an overestimation of DNA methylation levels [34]. A previous study reported lower overall methylation levels in long-read sequencing (Oxford Nanopore) compared to bisulfite-sequenced samples, attributing these differences to challenges in precisely aligning short-read sequences to the reference genome [9].

The methylation probabilities correlated strongly between the two methods for overlapping CpG sites, particularly in the autosomes (Pearson's r ≥ 0.78). Based on our results, differences in correlation at the primary level (Sequence-based) appear to influence the correlation at the secondary level (Functional Segment-based). We observed a stronger correlation between the two approaches in GC-rich regions and areas with higher GC densities. Notably, the functional genetic regions like promoters and coding sequences, which exhibit higher GC content, showed stronger correlation compared to intronic and intergenic regions. The correlation between the two detection methods may be influenced by the density of CpG sites, as well as methylation stability and conservation. GC-rich regions or areas with a high frequency of CpG dinucleotides are associated with important regulatory elements and high conservation [35]. We postulated that multiple consecutive mCs may enhance methylation detection in HiFi WGS by amplifying polymerase kinetic signals. Additionally, bisulfite sequencing, which tracks methylation through sequence composition changes, may also benefit from consecutive CpGs by making pattern detection easier. This may lead to greater consistency between detection methods. As a result, CpG shores and CpG shelves, which have lower CpG content and more variable methylation pattern compared to CpG islands, tend to show reduced correlation due to the different detection competencies of the two technologies. Functional genetic regions may also be impacted by the stronger correlation found in GC-rich areas. Promoters are CpG-rich, and methylation levels are highly conserved in these regions [35], leading to the highest correlation of methylation levels between the two detection methods in our findings. UTRs, which are often located near promoters, can share

similar environments. These are followed by exons with moderate CpG density and moderate variability, then introns and intergenic regions with low CpG density, more variable and context-dependent methylation [36–38]. Regulatory regions, including enhancers and open chromatin, are typically located in moderate to GC-rich genomic contexts [39–41]. In our data, these regions exhibited more stable methylation and higher concordance between platforms than low-GC or variable regions, suggesting that their GC content and functional conservation support more reliable methylation detection.

When considering repetitive and non-repetitive regions, the highest correlation was found in non-repetitive regions. Non-repetitive regions consist of unique and non-transposable sequences, making alignment and mapping easier. This clarity allows methylation patterns to be identified more precisely, leading to higher correlation between two detection methods. Tandem repeats consist of short, repeating nucleotide motifs, which can pose challenges for sequencing technologies to differentiate between methylated and unmethylated repeats [42]. LINEs and SINEs, as transposable elements with longer and more complex repetitive elements, present unique challenges for accurate alignment due to their repetitive nature [43]. The methylation patterns tend to be more diffuse and complex. Moreover, the sequence composition of these elements is typically AT-rich [44], which may hinder methylation detection. These factors may introduce variability in methylation detection across methods. Furthermore, we notice the correlation decline for chromosomes X and Y, and these chromosomes are attributed to overlapped CpG sites originating mainly from low GC densities and intergenic regions.

Additionally, to account for the prevalence of low-depth CpGs in WGBS, concordance with HiFi WGS was assessed using coverage-matched sites. Concordance increased with coverage, with high-depth CpGs showing strong agreement, while low-depth sites contributed to reduced correlation. A slight decline at the highest-depth bins may reflect increased variability and limited number of matched sites in these extreme ranges. Building on this, we performed site-level, depth-matched subsampling to equalize coverage between platforms. The overall correlation remained robust, which may be partly due to the effective reduction in read depth introduced by the down-sampling process. The similarity between the down-sampled (Fig 8) and original datasets (Figs 4 and 5) suggests that although absolute coverage influences signal strength, the underlying methylation trends remain robust across technologies.

When we further visualized the distribution of Pearson correlation coefficients across 1,000 depth-matched downsampling replicates (S24 Fig), a nonlinear relationship between read depth and inter-platform concordance was observed. Specifically, correlation values initially decreased from 4× to around 20×. At 4–5×, this is likely due to stochastic noise inherent to low-coverage methylation calling, where fractional methylation values are constrained to coarse levels (e.g., 0%, 25%, 50%, 75%, 100%) and therefore have a higher chance of coincidentally matching between platforms, limiting resolution. At 6–20×, predictions may still be influenced by stochastic variability, greater variability in methylation percentage estimates, or insufficient signal accumulation—particularly in regions with low sequence complexity or partial methylation. As coverage increased from 21× to 46×, correlation improved, reflecting greater reliability and precision in methylation estimates as more reads contribute to the aggregated signal. Interestingly, correlation declined again beyond 46×, a pattern we attribute to the small number of CpG sites achieving such high coverage, which introduces sampling noise and reduces statistical robustness. Based on these results, we recommend a minimum coverage of approximately 20× per CpG site for confident methylation quantification using HiFi WGS, balancing both resolution and genomic breadth.

Our study outcomes from both methylation detection methods are consistent with established biological principles. Methylation can vary across different regions of the genome, leading to diverse effects on gene activities in various genomic regions [1]. In this study, CpG islands had the lowest frequency of mCs and methylation level, followed by CpG shores and CpG shelves. CpG islands are generally unmethylated, and frequently serve as promoters for housekeeping genes [45]. Moreover, mCs tend to locate farther from these CpG islands [46]. This pattern is not unique to individuals with DS and has been consistently observed in healthy populations. Although methylation levels can be influenced by age, promoter-associated CpG islands generally remain unmethylated throughout life. Prior studies have shown that in DS, CpG islands largely retain this unmethylated status, with disease-associated methylation changes occurring in a gene-specific rather than global manner —for example, differential methylation of transcription factor genes (e.g., ZNF

and HOX families) and CpG islands within the HIST1 gene cluster has been observed in DS patients, potentially affecting chromatin remodeling [47]. These observations suggest that the reduced methylation levels observed in CpG islands in our DS twin samples reflect general epigenomic architecture rather than disease- or age-specific disruption. In 5-base windows, the presence of mCs decreases in regions with 100 percent GC density due to the high prevalence of CpG Islands and promoters (S21 Fig). Among genic regions, promoters that are CG-rich stand out with the least occurrence of mCs and lowest methylation level [48]. Unmethylated DNA in these promoters enables DNA to modify a structure that disrupts nucleosomes and enhances the attachment of essential factors for transcription initiation [48]. Gene bodies, particularly in intronic regions, often exhibit higher DNA methylation levels in actively transcribed genes. This DNA methylation can modify histones and alter chromatin structure, ultimately increasing transcription activity [48–51].

Additionally, our results show that mCs in tandem repetitive regions were the lowest when compared to other repetitive regions including non-repetitive regions. The proportion of CpG sites with low read depth was highest in tandem repeat regions for WGBS, which likely contributes to the reduced number of detected mCs in this region. Tandem repeats may pose challenges in methylation detection due to limitations in sequencing technologies, mapping, or methylation detection algorithms [9]. These regions often have low DNA entropy—a measure of sequence complexity—due to their repetitive and homogeneous sequence content, making them particularly difficult for short-read mapping. In our study, we calculated Shannon entropy values across genomic regions and found that tandem repeats had the lowest average entropy (1.36), compared to 1.96 for interspersed repeats (LINEs and SINEs) and 1.90 for non-repetitive regions (see Methods). Moreover, we found slightly higher methylation levels in repetitive regions compared to non-repetitive regions, with SINEs showing the highest levels. Repetitive elements are typically methylated to maintain a heterochromatic state [52]. Hypomethylation in transposable or interspersed repetitive elements contributes to genetic instability, making DNA methylation crucial for keeping them silenced [53]. Methylation profiles aligned to gene structures in this study also revealed a consistent pattern with known epigenetic regulation: hypomethylation near the TSS, increased methylation across gene bodies [54].

Recently, a previous study performed the initial systematic comparison of mC detection tools for long-read sequencing [9]. Our results from HiFi WGS and WGBS comparison in DS samples, despite being based on a limited number of samples, align with the findings of that study. Our findings aligned with previous studies in several key aspects: a strong correlation was observed between the two methylation detection methods, methylation probabilities consistently avoided extreme values of 0 or 1 in HiFi WGS, and greater CpG site detection was observed in HiFi WGS. The previous study found a higher correlation (0.97 compared to our 0.80) between HiFi WGS and WGBS, possibly due to their method of averaging 5-mCpG rates across all individuals, rather than performing a sample-matched comparison like ours. Furthermore, both our study and the previous study align with biological principles. The previous study observed a lack of mC within 50 bp intervals relative to the TSSs, consistent with our findings in promoter regions. Therefore, the results of this study were consistent with the outcomes of previous research, despite the presence of an existing medical condition (DS). Moreover, this study expands the comparison to various genomic regions across the genome. We used an updated 5mC prediction tool for HiFi WGS (Jasmine instead of Primrose), and the results remain consistent with those of the prior study.

We performed this analysis using monozygotic twins with DS, who were part of a larger ongoing rare disease study focused on identifying genomic variants associated with epilepsy, congenital heart disease, and DS. These samples were originally collected for comprehensive genome-wide analysis, including SNVs, SVs, CNVs, STRs, and DNA methylation. Our comparison confirms that the agreement between HiFi and WGBS methylation measurements is robust even in the presence of trisomy 21. This extends prior comparisons conducted in healthy individuals to a clinical setting and supports the broader applicability of long-read methylation profiling in disease-focused genomic research.

A key limitation of this study is the small sample size, as the analysis was performed on a single pair of monozygotic twins with Down syndrome. While this provided a unique opportunity to evaluate methylation concordance using matched short- and long-read data in a controlled genetic background, the findings may not fully capture variability across

individuals or disease states. Future studies with larger and more diverse cohorts will be important to confirm the generalizability of our observations and further validate the robustness of HiFi methylation profiling across biological contexts.

## Conclusions

This study assessed the concordance of CpG methylation patterns between HiFi WGS and traditional WGBS in a pair of monozygotic twins with DS, across both sequence-based and functional genomic contexts. Our results demonstrate strong agreement in methylation levels between the two technologies, particularly in GC-rich and biologically relevant regions. HiFi WGS identified more mCs overall, especially in repetitive regions such as tandem repeats, where WGBS detection was limited by lower sequencing depth. In contrast, WGBS tended to report higher average methylation levels across shared CpG sites. Based on coverage-matched and down-sampling analyses, we recommend a minimum per-site coverage of 20× to achieve reliable and concordant methylation estimates. While HiFi WGS offers enhanced coverage in challenging genomic regions, both technologies yield biologically consistent methylation profiles. Combined with prior studies in individuals without known genetic conditions, our findings suggest that WGBS and HiFi WGS are concordant and robust for genome-wide methylation profiling across both typical and complex genetic backgrounds such as DS.

## Supporting information

**S1 Table. Overview of sequencing results from WGBS (wg-blimp) and HiFi WGS data.**
(PDF)

**S2 Table. Comparison of WGBS methylation and bisulfite conversion metrics between wg-blimp and Bismark pipelines.**
(PDF)

**S3 Table. Comparison of CpG methylation statistics between Bismark (WGBS) and HiFi WGS.**
(PDF)

**S4 Table. Comparison of mC detection at 50% and 80% methylation thresholds across platforms.**
(PDF)

**S1 Fig. CpG site detection across increasing coverage thresholds and cumulative coverage.** The plots illustrate how the number of detected CpG sites varies with increasing coverage thresholds for (A) HiFi WGS, (B) WGBS (wg-blimp), (C) WGBS (Bismark).
(PDF)

**S2 Fig. Distribution of mCs (≥50% methylation) across primary (sequence-level) genomic contexts in HiFi WGS and WGBS (Bismark).** Proportions of mCs (defined as ≥50% methylation with ≥4 × read coverage) are shown across sequence-based features: (A) CpG regions (islands, shores, and shelves), (B) CG density categories, and (C) repetitive elements. Data are shown for HiFi WGS, Bismark, overlapping mCs (Overlap), uniquely identified mCs in HiFi WGS (Unique to HiFi WGS), uniquely identified in Bismark (Unique to Bismark), and the difference between the unique sets (Δ unique sites: HiFi WGS vs. Bismark).
(PDF)

**S3 Fig. Distribution of mCs (≥50% methylation) across secondary (functional level) genomic contexts in HiFi WGS and WGBS (Bismark).** Methylated CpG proportions (≥50% methylation with ≥4 × read coverage) by: (A) gene-associated regions, (B) regulatory elements (open chromatin and enhancers), and (C) chromosomes. Data are presented for HiFi WGS, Bismark, Overlap, Unique to HiFi WGS, Unique to Bismark, and Δ unique sites.
(PDF)

**S4 Fig. Distribution of mCs (≥50% methylation) across primary (sequence-level) genomic contexts in WGBS (Bismark) and WGBS (wg-blimp).** Proportions of mCs (defined as ≥50% methylation with ≥4 × read coverage) are shown across sequence-based features: (A) CpG regions (islands, shores, and shelves), (B) CG density categories, and (C) repetitive elements. Data are shown for Bismark, wg-blimp, overlapping mCs (Overlap), uniquely identified mCs in Bismark (Unique to Bismark), uniquely identified in wg-blimp (Unique to wg-blimp), and the difference between the unique sets (Δ unique sites: Bismark vs. WGBS).
(PDF)

**S5 Fig. Distribution of mCs (≥50% methylation) across secondary (functional level) genomic contexts in WGBS (Bismark) and WGBS (wg-blimp).** Methylated CpG proportions (≥50% methylation with ≥4 × read coverage) by: (A) gene-associated regions, (B) regulatory elements (open chromatin and enhancers), and (C) chromosomes. Data are presented for Bismark, wg-blimp, Overlap, Unique to Bismark, Unique to wg-blimp, and Δ unique sites.
(PDF)

**S6 Fig. Distribution of mCs (≥80% methylation) across primary (sequence-level) genomic contexts in HiFi WGS and WGBS (wg-blimp).** Proportions of mCs (defined as ≥80% methylation with ≥4 × read coverage) are shown across sequence-based features: (A) CpG regions (islands, shores, and shelves), (B) CG density categories, and (C) repetitive elements. Data are shown for WGBS, HiFi WGS, overlapping mCs (Overlap), uniquely identified mCs in WGBS (Unique to WGBS), uniquely identified in HiFi WGS (Unique to HiFi WGS), and the difference between the unique sets (Δ unique sites: HiFi WGS vs. WGBS).
(PDF)

**S7 Fig. Distribution of mCs (≥80% methylation) across secondary (functional level) genomic contexts in HiFi WGS and WGBS (wg-blimp).** Methylated CpG proportions (≥80% methylation with ≥4 × read coverage) by: (A) gene-associated regions, (B) regulatory elements (open chromatin and enhancers), and (C) chromosomes. Data are presented for WGBS, HiFi WGS, Overlap, Unique to WGBS, Unique to HiFi WGS, and Δ unique sites.
(PDF)

**S8 Fig. Distribution of mCs (≥80% methylation) across primary (sequence-level) genomic contexts in WGBS (Bismark) and WGBS (wg-blimp).** Proportions of mCs (≥80% methylation with ≥4 × read coverage) are shown across sequence-based features: (A) CpG regions (islands, shores, and shelves), (B) CG density categories, and (C) repetitive elements. Data are shown for Bismark, wg-blimp, overlapping mCs (Overlap), uniquely identified mCs in Bismark (Unique to Bismark), uniquely identified in wg-blimp (Unique to wg-blimp), and the difference between the unique sets (Δ unique sites: Bismark vs. WGBS).
(PDF)

**S9 Fig. Distribution of mCs (≥80% methylation) across secondary (functional level) genomic contexts in WGBS (Bismark) and WGBS (wg-blimp).** Methylated CpG proportions (≥80% methylation with ≥4 × read coverage) by: (A) gene-associated regions, (B) regulatory elements (open chromatin and enhancers), and (C) chromosomes. Data are presented for Bismark, wg-blimp, Overlap, Unique to Bismark, Unique to wg-blimp, and Δ unique sites.
(PDF)

**S10 Fig. Distribution of mCs (≥80% methylation) across primary (sequence-level) genomic contexts in HiFi WGS and WGBS (Bismark).** Proportions of mCs (defined as ≥80% methylation with ≥4 × read coverage) are shown across sequence-based features: (A) CpG regions (islands, shores, and shelves), (B) CG density categories, and (C) repetitive elements. Data are shown for HiFi WGS, Bismark, overlapping mCs (Overlap), uniquely identified mCs in HiFi WGS (Unique to HiFi WGS), uniquely identified in Bismark (Unique to Bismark), and the difference between the unique sets (Δ unique sites: HiFi WGS vs. Bismark).
(PDF)

**S11 Fig. Distribution of mCs (≥80% methylation) across secondary (functional level) genomic contexts in HiFi WGS and WGBS (Bismark).** Methylated CpG proportions (≥80% methylation with ≥4×read coverage) by: (A) gene-associated regions, (B) regulatory elements (open chromatin and enhancers), and (C) chromosomes. Data are presented for HiFi WGS, Bismark, Overlap, Unique to HiFi WGS, Unique to Bismark, and Δ unique sites.
(PDF)

**S12 Fig. Examination of mC position loss in each method.** Proportion of CpGs from WGBS at corresponding positions to uniquely mC positions detected by HiFi in twins A (WGBS from unique HiFi A) and B (WGBS from unique HiFi B), and CpGs from HiFi at corresponding positions to uniquely mC positions detected by WGBS of twin A (HiFi from unique WGBS A) and B (HiFi from unique WGBS B), considering alternative variants (pink), sequencing depth <4 (blue), and methylation levels < 50 (violet).
(PDF)

**S13 Fig. Proportion of low-coverage CpG Sites in various genomic contexts.** The proportion of low depth coverage CpGs positions from WGBS at corresponding positions to uniquely mC positions detected by HiFi in twins A (WGBS from unique HiFi A) and twin B (WGBS from unique HiFi B), and CpGs from HIFI at corresponding positions to uniquely mC positions detected by WGBS of twin A (HiFi from unique WGBS A) and twin B (HiFi from unique WGBS B), considering by (A) CpG regions, (B) GC densities, (C) Repeatitive/ non-repeaitive regions, (D) genetic regions and (E) chromosomes.
(PDF)

**S14 Fig. Methylation levels and correlation across primary (sequence-level) genomic contexts in HiFi WGS and WGBS (Bismark).** Methylation levels (methylation probabilities) and Pearson correlation between WGBS and HiFi WGS across: (A) CpG regions (CpG islands, shores, and shelves), (B) CG density categories, and (C) repetitive elements.
(PDF)

**S15 Fig. Methylation levels and correlation across secondary (functional level) genomic contexts in HiFi WGS and WGBS (Bismark).** Methylation levels and Pearson correlation between WGBS and HiFi WGS across: (A) gene-associated regions, (B) regulatory regions (open chromatin and enhancers), and (C) chromosomes.
(PDF)

**S16 Fig. Methylation levels and correlation across primary (sequence-level) genomic contexts in WGBS (Bismark) and WGBS (wg-blimp).** Methylation levels (methylation probabilities) and Pearson correlation between WGBS and HiFi WGS across: (A) CpG regions (CpG islands, shores, and shelves), (B) CG density categories, and (C) repetitive elements.
(PDF)

**S17 Fig. Methylation levels and correlation across secondary (functional level) genomic contexts in WGBS (Bismark) and WGBS (wg-blimp).** Methylation levels and Pearson correlation between WGBS and HiFi WGS across: (A) gene-associated regions, (B) regulatory regions (open chromatin and enhancers), and (C) chromosomes.
(PDF)

**S18 Fig. Methylation signal distribution relative to gene structure.** Average methylation levels are plotted relative to gene structures, including 2 kb upstream of the transcription start site (TSS), the gene body (scaled to uniform length), and 2 kb downstream of the transcription end site (TES). The plot compares methylation patterns in promoter, gene body, and downstream regions across platforms (HiFi WGS, Bismark, and wg-blimp (MethylDackel).
(PDF)

**S19 Fig. Methylation levels of non-CpG sites (CHG and CHH) across genomic contexts in WGBS (wg-blimp), and WGBS (Bismark).** Comparisons are shown across: (A) CpG-related regions (islands, shores, and shelves), (B) CG density categories, (C) repetitive elements, (D) gene-associated regions, and (E) regulatory regions (open chromatin and enhancers).
(PDF)

**S20 Fig. Two-dimensional heatmaps with best-fit lines showing methylation concordance between WGBS (wg-blimp) and HiFi WGS across genomic contexts (Twin B).** Scatter plots (2D binned heatmaps) with linear regression lines illustrate methylation level concordance between WGBS and HiFi for CpG sites across: (A) CpG regions (islands, shores, and shelves), CG density categories, repetitive elements, (B) gene-associated regions, and regulatory regions (open chromatin and enhancers).
(PDF)

**S21 Fig. Distribution of genetic regions across different GC density bins.** Proportions of genetic regions (promoters, exons, introns, UTRs, intergenic regions) categorized by GC density levels (20, 40, 60, 80, and 100%).
(PDF)

**S22 Fig. Genomic context distribution across chromosomes.** Proportion of (A) CpG regions, (B) GC density categories, and (C) genetic regions across each chromosome based on overlapping CpG positions identified by WGBS (wg-blimp) and HiFi WGS in the twin samples.
(PDF)

**S23 Fig. Methylation concordance between HiFi WGS and WGBS after depth-matched subsampling (Twin B).** Methylated and unmethylated counts were randomly downsampled at each CpG site to the lower read depth between platforms, ensuring matched coverage for fair comparison. Pearson correlation coefficients (r) of methylation levels were calculated over 1,000 subsampling iterations. (A) Boxplots of correlation coefficients across genomic contexts. (B) Boxplots of correlation coefficients stratified by depth bins.
(PDF)

**S24 Fig. Methylation concordance between HiFi WGS and WGBS across depth bins after depth-matched downsampling.** Downsampling was performed as described in Fig 8. CpG sites were stratified by coverage depth bins. Boxplots show the distribution of Pearson correlation coefficients (r) across 1,000 iterations for each bin. (A) Twin A. (B) Twin B.
(PDF)

## Acknowledgments

The Scholarship from the Graduate School, Chulalongkorn University to commemorate the 72nd anniversary of his Majesty King Bhumibol Aduladej is gratefully acknowledged.

## Author contributions

**Conceptualization:** Monnat Pongpanich.

**Formal analysis:** Kanyanee Promsawan, Monnat Pongpanich.

**Funding acquisition:** Vorasuk Shotelersuk.

**Methodology:** Kanyanee Promsawan, Chalurmpon Srichomthong.

**Software:** Kanyanee Promsawan.

**Supervision:** Monnat Pongpanich, Vorasuk Shotelersuk.

**Visualization:** Kanyanee Promsawan.

**Writing – original draft:** Kanyanee Promsawan, Chalurmpon Srichomthong.

**Writing – review & editing:** Monnat Pongpanich, Vorasuk Shotelersuk.

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
