## [Decision Letter · Decision Letter 0]

1 Apr 2025

Dear Dr. Pongpanich,

Thank you for submitting your manuscript to PLOS ONE. After careful consideration, we feel that it has merit but does not fully meet PLOS ONE’s publication criteria as it currently stands. Therefore, we invite you to submit a revised version of the manuscript that addresses the points raised during the review process.

The stated goal of the project – to compare methylation calling between traditional WGBS and PacBio HiFi sequencing – is certainly valuable. The authors assert that comparing methylation calling between WGBS and PacBio HiFi sequencing is particularly pertinent for complex conditions like Down syndrome. The authors acknowledge that their experiments were not designed for direct comparison across factors such as read depths, sample sizes, or conditions. While they emphasize that the ‘available data’ presented a unique opportunity to assess cross-platform consistency, this approach raises important concerns. To strengthen the reliability of their conclusions, it is essential that the authors address the analytical and methodological concerns raised by the reviewers.

We look forward to receiving your revised manuscript.

Kind regards,

Purnima Singh, PhD

Academic Editor

PLOS ONE

3. For studies involving third-party data, we encourage authors to share any data specific to their analyses that they can legally distribute. PLOS recognizes, however, that authors may be using third-party data they do not have the rights to share. When third-party data cannot be publicly shared, authors must provide all information necessary for interested researchers to apply to gain access to the data. (https://journals.plos.org/plosone/s/data-availability#loc-acceptable-data-access-restrictions)

4) All necessary contact information others would need to apply to gain access to the data.

Additional Editor Comments:

The stated goal of the project – to compare methylation calling between traditional WGBS and PacBio HiFi sequencing – is certainly valuable. The authors assert that comparing methylation calling between WGBS and PacBio HiFi sequencing is particularly pertinent for complex conditions like Down syndrome. The authors acknowledge that their experiments were not designed for direct comparison across factors such as read depths, sample sizes, or conditions. While they emphasize that the ‘available data’ presented a unique opportunity to assess cross-platform consistency, this approach raises important concerns. To strengthen the reliability of their conclusions, it is essential that the authors address the analytical and methodological concerns raised by the reviewers.

Reviewers' comments:

Reviewer's Responses to Questions

**Comments to the Author**

1. Is the manuscript technically sound, and do the data support the conclusions?

Reviewer #1: Yes

Reviewer #2: Yes

Reviewer #3: No

2. Has the statistical analysis been performed appropriately and rigorously?

Reviewer #1: No

Reviewer #2: Yes

Reviewer #3: N/A

3. Have the authors made all data underlying the findings in their manuscript fully available?

Reviewer #1: No

Reviewer #2: Yes

Reviewer #3: No

4. Is the manuscript presented in an intelligible fashion and written in standard English?

Reviewer #1: Yes

Reviewer #2: Yes

Reviewer #3: Yes

Reviewer #1: In this manuscript, the authors assessed CpG methylation concordance between WGBS and HiFi sequencing using monozygotic twins. They found high correlation in CpG methylation results from the two technologies, while also some differences. While the results are interesting, some additional analysis will be beneficial to reveal more results to the readers.

Major points:

1. More QC metrics for either data types, e.g. # of CpGs with different coverage cutoff, % of methylated Cs under CpG, CHH, CHG context, QC for bisulfite conversion efficiency estimated by mC outside of CpG context, etc.

2. Different depth might bias the results. It will be ideal if the author can down-sample the HiFi data to match the WGBS depth;

3. To reveal how depth cutoff affects the correlation, scatter plot between the two technologies colored by coverage in WGBS should be generated.

Minor points:

4. To avoid any bias in results of WGBS, another tool such as Bismark should be used to process the data and show no substantial difference from current method.

5. Annotations with enhancers or open chromatin regions will be helpful.

6. Average signal plot around gene body + promoter will be helpful.

7. How about CHH and CHG context? Can authors also report those regions?

8. It will be interesting to perform comparison between the twins to identify differential mCpGs and compare the findings from either technologies to see if any bias present.

Reviewer #2: I enjoyed reading this manuscript that aims at contracting the methylation landscape between HiFi and WGBS in Down Syndrome patients. While the manuscript was a nice read it needs some additional work to become ready for publication.

Line 45: “pivotal epigenetic genome modification”, remove the word “genome” and let the sentence be “pivotal epigenetic modification”.

Materials and Methods: Provide information about the twins, are they male/female, age at recruitment, body composition (height and weight). It can be concluded from your results that they are males because of Y chromosome data, but you still need to say in the method section that they are males.

Line 115: “the recruitment period spanning from July 18, 2019, to July 17, 2025.” The date is incorrect. We are not in July 2025 yet. Also, if you recruited only a single pair of twins why was it done over a long period?

Line 246 – 248: you say that CpG islands show “demonstrated limited methylated CpG sites”. Is this something commonly reported in Down syndrome cases where CpG islands are not densely methylated? If yes, say something about that and provide a reference. In healthy people it is typical to see unmethylated islands, would that be the case in DS? Also we don’t know the age of the twins, if they are young, then the lack of methylation could be to age and not any other reason.

Line 269: “there were a depth of coverage less than” change it so it says: there was a depth of coverage less than”.

Result section: For all correlation combinations, you need to provide scatter plots to show correlation. In those plots, you need to show r value and line of best fit.

In supp fig 3 and 4, the axes should be labeled. Indicate what the x axis is and what the y axis is.

Line 419: This is the first time entropy is mentioned in the manuscript. How was entropy measured? There is no mention of this in the method section.

Discussion

It is not clear what role DS plays in this research. Was it expected that DS would be a major disruptor of methylome? The main purpose of the study is to highlight the differences between HiFi and WGBS. This being done in DS cases is not clear. Was there a reason to think that DS would lead to different methylation readings? Based on the results, the differences between the two methods is based on technology and methodology, and DS plays no role in this difference.

The authors did not show any concern with the sample size. You only have one pair of DS twins. You need to list that as a limitation.

Reviewer #3: The stated goal of this project - to compare methylation calling via traditional WGBS with PacBio HiFi sequencing - is a worthwhile undertaking. However, I’m not sure the paper in its current form accomplishes that goal.

The authors argue that a comparison of methylation calling from WGBS and PacBio HiFi sequencing is especially relevant in “complex conditions like Down syndrome”. They state in the Introduction that there “... are still limited studies on the concordance of 5-mC detection across various genomic regions, especially in the context of genetic conditions like Down syndrome.”

From these statements, it seems there are three distinct goals: (1) to compare the two platforms in terms of calling DNA methylation, (2) to compare reproducibility of methylation calls in different genomic contexts, and (3) to study DNA methylation in Down syndrome. To accomplish any of these individual goals would likely require a unique study design. By trying to do many things at once, the current manuscript sacrifices the ability to do any one task sufficiently.

If the goal is to study the role of DNA methylation in down syndrome, the choice of whole blood samples is not well motivated. DNA methylation patterns are cell type specific. It’s not clear how blood cell phenotypes are related to Down syndrome.

If the goal is to compare reproducibility of methylation calls in different genomic contexts, there needs to be a more careful stratification of the genome. We expect that DNA methylation will mark most CpGs in the genome, with the exception being CpG islands, which mark the promoters of most human genes (see e.g. DOI 10.1186/s13072-017-0130-8 10.1038/s41580-019-0159-6). Where the differences in HiFi PacBio sequencing vs WGBS may be most apparent is in highly repetitive regions, such as transposons or pericentromeric regions. A more thorough analysis of this would potentially be informative.

If the goal is to study reproducibility of methylation calls, it is important to control for coverage across the genome. The different coverages for WGBS and HiFi in Table 1 will necessarily lead to differences in methylation calls. The dramatically different read lengths will also introduce bias in methylation interrogation. If the authors want to compare WGBS and HiFi in an unbiased manner, they could sample from the HiFi library to achieve coverage and read lengths analogous to the WGBS libraries and repeat their analysis. In the first paragraph of the results section the authors add that the sequencing depth is not controlled because of “reliance on existing data”. Are these datasets all published?

In addition to these conceptual concerns, I have technical concerns as well:

The wg-blimp analysis pipeline used for the WGBS analysis is not well defended. Utilizing a more standard approach (e.g. Methpipe?) would be more appropriate. One puzzling aspect of the wg-blimp pipeline is the prioritization of quality control _after_ mapping and de-duplication.

There are several seemingly arbitrary thresholds introduced in the manuscript. For example, regarding CpGs as methylated with methylation level is >= 50 with coverage >= 4 is not justified. Generally DNA methylation profiles follow a bimodal distribution with the largest mode corresponding to CpGs being methylated and another mode corresponding to unmethylated. Do the methylation values for HiFi and WGBS here follow that pattern? That could be used to set thresholds.

**Do you want your identity to be public for this peer review?** For information about this choice, including consent withdrawal, please see our Privacy Policy

Reviewer #1: No

Reviewer #2: No

Reviewer #3: No

---

## [Author Response · Author response to Decision Letter 1]

30 Jun 2025

Dear Editor,

We sincerely appreciate the reviewers’ thoughtful comments and constructive suggestions, which have greatly helped us improve the clarity and overall quality of our manuscript. We have carefully addressed each point raised and revised the manuscript accordingly. Please find our detailed point-by-point responses below, with corresponding line numbers from the revised manuscript.

Reviewer #1: In this manuscript, the authors assessed CpG methylation concordance between WGBS and HiFi sequencing using monozygotic twins. They found high correlation in CpG methylation results from the two technologies, while also some differences. While the results are interesting, some additional analysis will be beneficial to reveal more results to the readers.

Major points:

1. More QC metrics for either data types, e.g. # of CpGs with different coverage cutoff, % of methylated Cs under CpG, CHH, CHG context, QC for bisulfite conversion efficiency estimated by mC outside of CpG context, etc.

Reply: We thank the reviewer for the suggestions regarding additional quality control metrics. In response, we have expanded our QC reporting as follows:

- We now report the number of CpG sites at each coverage depth for both platforms in Supplementary Figure S1.

- For the WGBS dataset, we provide the percentage of methylated cytosines in CpG, CHG, and CHH contexts in Supplementary Table 2.

- We evaluated bisulfite conversion efficiency by measuring the proportion of methylated cytosines in non-CpG (CHH) contexts, which serves as an estimate of conversion failure, and report this in Supplementary Table 2.

However, for the PacBio HiFi long-read methylation calls, current tools do not support confident context-specific calling in CHH or CHG contexts. Therefore, we restrict CHH/CHG analysis to WGBS only, as noted in the revised manuscript.

We have updated the manuscript accordingly in lines 262-266:

“Bisulfite conversion efficiency was consistently high (97.2–97.36%) across samples and pipelines, confirming effective conversion of unmethylated cytosines. Correspondingly, CpG methylation levels were high (83.7–85.4%), while non-CpG methylation remained low, with CHG and CHH methylation each accounting for only 2.4–2.8% (Supplementary Table 2).”

And in lines 288-301:

“To further assess CpG detection sensitivity, we analyzed CpG site counts across increasing minimum read coverage thresholds (4× to >60×) and plotted cumulative coverage distributions across three datasets: HiFi WGS, wg-blimp, and Bismark. The depth distribution for PacBio HiFi (Supplementary Fig S1A) shows a unimodal and symmetric pattern peaking at 28–30×, indicating relatively uniform coverage. In contrast, both WGBS datasets (Supplementary Fig S1B,C) display right-skewed distributions, with the majority of CpGs covered at low depth (4–10×) and relatively few achieving higher coverage. Over 90% of CpGs in the PacBio HiFi dataset have ≥10× coverage, compared to approximately 65% in the wg-blimp WGBS dataset and under 50% in the Bismark WGBS dataset, as estimated from the cumulative plots (Supplementary Fig S1). Notably, while most CpGs in the WGBS datasets are concentrated at lower depths, the final bin (>60×) includes a small number of CpG sites with extremely high coverage (some exceeding 4000×), which inflate the average coverage values reported in the summary tables and explain the discrepancy between the mean and the more typical coverage levels observed.”

2. Different depth might bias the results. It will be ideal if the author can down-sample the HiFi data to match the WGBS depth;

Reply: We thank the reviewer for pointing out the potential confounding effect of coverage differences between platforms. To address this concern, we initially explored read-level downsampling of PacBio HiFi BAM files. However, this approach proved infeasible due to the nature of long-read sequencing: a single HiFi read typically spans multiple CpG sites, so removing a read to reduce depth at one position would inadvertently affect neighboring sites as well. As a result, it is not practical to control read depth independently at the CpG level using read-based downsampling.

Instead, we implemented a site-level depth-matched subsampling strategy, where we randomly sampled methylated and unmethylated counts at each CpG site. The number of sampled reads was matched to the minimum read depth between PacBio HiFi and WGBS at that site, ensuring fair comparison. This method preserved per-site resolution while eliminating depth as a confounding factor.

We repeated the subsampling process for 1,000 independent replicates across depth levels from 4× to > 60×, and computed Pearson correlation coefficients (r) between platforms in each replicate. The results are shown in Figure 8 and Supplementary Figures S24.

We have updated the Results section accordingly in lines 522–538:

“To further account for coverage-related bias, we implemented a site-level, depth-matched down-sampling strategy for the HiFi WGS and WGBS datasets. At each CpG site, methylation values from the higher-depth platform were randomly down-sampled to match the read depth of the lower-depth platform. This approach enabled the inclusion of a much larger number of CpG sites compared to equal-depth filtering (Fig 7; Supplementary Figure S23). To reduce stochastic variation from random sampling, we performed 1,000 rounds of down-sampling per CpG site and used the average methylation level for downstream analysis. Pearson correlation coefficients were then calculated between the down-sampled HiFi WGS and WGBS methylation levels across various genomic features. The results demonstrated good concordance, with correlation patterns (Fig. 8) that closely resembled those observed in the original (non-downsampled) data (Fig. 4, 5). To further examine the influence of sequencing depth on methylation concordance, we visualized the distribution of Pearson correlation coefficients across 1,000 down-sampling replicates at each PacBio depth level (Supplementary Fig. S24). This analysis provides a complementary view to Figure 8 and reveals a non-linear relationship between depth and concordance. Specifically, correlation values initially decline as depth increases from 4× to approximately 20×, then gradually rise and reach a peak at 46×. Beyond this point, correlation values decline again.”

We also revised the Discussion (lines 635–656) to reflect these insights:

“Building on this, we performed site-level, depth-matched subsampling to equalize coverage between platforms. The overall correlation remained robust, which may be partly due to the effective reduction in read depth introduced by the down-sampling process. The similarity between the down-sampled (Figure 8) and original datasets (Figures 4 and 5) suggests that although absolute coverage influences signal strength, the underlying methylation trends remain robust across technologies.

When we further visualized the distribution of Pearson correlation coefficients across 1,000 depth-matched downsampling replicates (Supplementary Fig. S24), a nonlinear relationship between read depth and inter-platform concordance was observed. Specifically, correlation values initially decreased from 4× to around 20×. At 4–5×, this is likely due to stochastic noise inherent to low-coverage methylation calling, where fractional methylation values are constrained to coarse levels (e.g., 0%, 25%, 50%, 75%, 100%) and therefore have a higher chance of coincidentally matching between platforms, limiting resolution. At 6–20×, predictions may still be influenced by stochastic variability, greater variability in methylation percentage estimates, or insufficient signal accumulation—particularly in regions with low sequence complexity or partial methylation. As coverage increased from 21× to 46×, correlation improved, reflecting greater reliability and precision in methylation estimates as more reads contribute to the aggregated signal. Interestingly, correlation declined again beyond 46×, a pattern we attribute to the small number of CpG sites achieving such high coverage, which introduces sampling noise and reduces statistical robustness. Based on these results, we recommend a minimum coverage of approximately 20× per CpG site for confident methylation quantification using HiFi WGS, balancing both resolution and genomic breadth.”

3. To reveal how depth cutoff affects the correlation, scatter plot between the two technologies colored by coverage in WGBS should be generated.

Reply: We thank the reviewer for this insightful suggestion. As recommended, we investigated the relationship between methylation concordance and WGBS coverage. Initially, we explored a scatter plot colored by WGBS depth; however, this approach resulted in severe overplotting due to the large number of CpG sites, making patterns difficult to interpret.

To address this, we opted to generate two-dimensional binned heatmaps (Figure 7 and Supplementary Figures S23), which allow for clear visualization of CpG density while preserving the correlation structure between WGBS and HiFi methylation levels. Importantly, we stratified the analysis by equal depth levels—restricting to CpG sites with matched coverage between platforms—and grouped them into non-overlapping bins (e.g., 5–10×, 11–15×, etc.).

We have updated the Results section accordingly in lines 631–635:

“Additionally, to account for the prevalence of low-depth CpGs in WGBS, concordance with HiFi WGS was assessed using coverage-matched sites. Concordance increased with coverage, with high-depth CpGs showing strong agreement, while low-depth sites contributed to reduced correlation. A slight decline at the highest-depth bins may reflect increased variability and limited number of matched sites in these extreme ranges.”

We also revised the Discussion (lines 512–522) to reflect these insights:

“To address potential coverage-related bias in methylation concordance, we compared HiFi WGS and WGBS using coverage-matched CpG sites. At lower coverage bins (5–20×), correlations were moderate (r = 0.53–0.65), whereas at higher depths (≥31×), concordance improved substantially (r ≥ 0.72). A decline in correlation was observed in the highest-depth bins (Fig 7 (Twin A) and Supplementary Figure S23 (Twin B)). Although the scatterplots may appear noisy at first glance, this visual impression is primarily driven by a small number of discordant points, as indicated by dense clusters of blue-colored (low-frequency) dots. In contrast, the majority of data points are concentrated along the upper-right diagonal, highlighted by the yellow gradient, indicating strong agreement in methylation levels among coverage-matched CpGs. These results underscore the importance of sufficient sequencing depth for reliable methylation quantification.”

Minor points:

4. To avoid any bias in results of WGBS, another tool such as Bismark should be used to process the data and show no substantial difference from current method.

Reply: We thank the reviewer for this helpful suggestion. To evaluate the robustness of our findings with respect to WGBS processing pipelines, we re-analyzed the data using Bismark, a widely used and well-established bisulfite sequencing analysis tool, as suggested. We then compared the methylation levels from Bismark to HiFi and wg-blimp-based pipeline. The results are shown in Supplementary Figure S2- S11 and Supplementary Table S3.

We have updated the Results section accordingly in lines 312–333:

“Overall, HiFi WGS detected a greater number of mCs while maintaining broadly consistent methylation patterns with WGBS results generated by both the wg-blimp and Bismark pipelines in both twins (Fig. 2–3, Supplementary Figures S2–S3). While the overall trends were preserved across methods—for example, increasing methylation levels from CpG islands to shores and shelves—there were noticeable differences in specific genomic contexts. In particular, methylation profiles between HiFi WGS and Bismark showed greater divergence compared to those between HiFi WGS and wg-blimp. For example, in repetitive elements, HiFi WGS showed higher methylation in SINEs than in LINEs, whereas Bismark displayed the opposite trend (Fig. 2–3, Supplementary Figures S2–S3). Between the two WGBS analysis pipelines, wg-blimp consistently reported a higher number of mCs than Bismark, yet both showed largely concordant patterns across genomic features (Supplementary Figures S4–S5). To further assess the robustness of our methylation threshold, we applied a more stringent cutoff of 80%. The results remained consistent, with over 80% of mCs overlapping those identified using the original 50% threshold (Supplementary Table 4). Overall methylation patterns remained similar across comparisons—HiFi WGS vs. wg-blimp, HiFi WGS vs. Bismark, and wg-blimp vs. Bismark— with the proportion of mCs uniformly decreasing by approximately 10–15% across different genomic regions for all three methods (Supplementary Figures S6–S11). The comparison between the 50% and 80% thresholds also showed consistent trends across HiFi WGS, wg-blimp, and Bismark in most genomic contexts. An exception was observed in CG density categories, where the HiFi WGS profile shifted from a clear decreasing trend under the 50% threshold to a more variable, non-monotonic pattern under the 80% threshold (Fig. 2–3, Supplementary Figures S2–S11).”

And in lines 370-391:

“The distribution patterns in secondary regions were largely consistent between the two techniques for both thresholds (50% and 80%) across various genetic regions including regulatory regions (enhancer and open chromatin) but exhibited variation across certain chromosomes (Fig 3; Supplementary Figure S3, S5, S7, S9, S11). In genetic regions, the lowest proportion of mCs were remarkably indicated in promoters. HiFi WGS remarkably identified more mCs in exons than WGBS (Fig 3A; Supplementary Figure S3A, S5A, S7A, S9A, S11AFig 2D). For regulatory regions, both technologies showed a consistent pattern in which enhancer regions had a slightly higher proportion of mCs compared to open chromatin regions (Fig 3B; Supplementary Figure S3B, S5B, S7B, S9B, S11B). Across all comparisons, the chromosomal distribution of mCs exhibited consistent patterns across detection methods and thresholds, with the exception of chromosomes 15–16, where HiFi WGS showed a slightly elevated proportion of methylated CpGs, diverging from the downward trend observed in both Bismark and wg-blimp (Fig 3C; Supplementary Figure S3C, S5C, S7C, S9C, S11C). HiFi WGS reported the highest mC proportions across all chromosomes, followed by wg-blimp and then Bismark, with all methods showing a pronounced reduction on chromosome Y. Wg-blimp and Bismark produced highly comparable mC levels and profiles across chromosomes. Notably, under the more stringent 80% threshold, the proportion of mCs declined uniformly across chromosomes for each method; however, the relative differences between platforms and the chromosome-specific patterns remained consistent with those observed at the 50% threshold. These findings demonstrate that despite varying detection thresholds and analytical pipelines, the chromosome-level methylation landscape is largely reproducible and biologically meaningful (Fig 3C; Supplementary Figure S3C, S5C, S7C, S9C, S11C).”

We also revised the Discussion (lines 562–563) to reflect these insights:

“Moreover, wg-blimp and Bismark from WGBS data show concordance in methylation pattern with very high correlation (r > 0.9) across genome”

And in lines 570-571:

“Moreover, Bismark tends to report lower CpG depths than wg-blimp, which may be due to differences in alignment stringency, read processing, and CpG counting strategies.”

These results confirm that our conclusions are not dependent on the WGBS processing pipeline, and that Bismark yields results consistent with those obtained from wg-bimp.

5. Annotations with enhancers or open chromatin regions will be helpful.

Answer: In response to the reviewer’s suggestion, we added a new panel (panel B) to Figures 3 and 5, as well as to Supplementary Figures S3, S5, S7, S9, S11, S15, and S17. We used publicly available DNase I hypersensitivity site annotations from ENCODE to identify open chromatin regions, then inte

---

## [Decision Letter · Decision Letter 1]

20 Jul 2025

A Comparison of DNA Methylation Detection between HiFi Sequencing and Whole Genome Bisulfite Sequencing in Monozygotic Twins with Down Syndrome

PONE-D-25-00547R1

Dear Dr. Pongpanich,

We’re pleased to inform you that your manuscript has been judged scientifically suitable for publication and will be formally accepted for publication once it meets all outstanding technical requirements.

Kind regards,

Purnima Singh, PhD

Academic Editor

PLOS ONE

Additional Editor Comments (optional):

Reviewers' comments:

Reviewer's Responses to Questions

**Comments to the Author**

Reviewer #1: All comments have been addressed

Reviewer #2: All comments have been addressed

2. Is the manuscript technically sound, and do the data support the conclusions?

Reviewer #1: Yes

Reviewer #2: Yes

3. Has the statistical analysis been performed appropriately and rigorously?

Reviewer #1: Yes

Reviewer #2: Yes

4. Have the authors made all data underlying the findings in their manuscript fully available?

Reviewer #1: No

Reviewer #2: Yes

5. Is the manuscript presented in an intelligible fashion and written in standard English?

Reviewer #1: Yes

Reviewer #2: Yes

Reviewer #1: All of my comments have been addressed by the authors in this revision. Although the differential methylation results will add value to the manuscript, I agree it is appropriate for another manuscript.

Reviewer #2: The authors have addressed all my points adequately. I wish them all the best and look forward to reading more of their work.

**Do you want your identity to be public for this peer review?** For information about this choice, including consent withdrawal, please see our Privacy Policy

Reviewer #1: No

Reviewer #2: No

---

## [Editor Report · Acceptance letter]

PONE-D-25-00547R1

PLOS ONE

Dear Dr. Pongpanich,

I'm pleased to inform you that your manuscript has been deemed suitable for publication in PLOS ONE. Congratulations! Your manuscript is now being handed over to our production team.

Kind regards,

on behalf of

Dr. Purnima Singh

Academic Editor

PLOS ONE